# MAESTRO : Adaptive Sparse Attention and Robust Learning for Multimodal Dynamic Time Series

**Payal Mohapatra** **Yueyuan Sui** **Akash Pandey** **Stephen Xia** **Qi Zhu**

Northwestern University
Evanston, IL, USA
{payal.mohapatra, qzhu}@northwestern.edu

## Abstract

From clinical healthcare to daily living, continuous sensor monitoring across multiple modalities has shown great promise for real-world intelligent decision-making but also faces various challenges. In this work, we introduce MAESTRO, a novel framework that overcomes key limitations of existing multimodal learning approaches: (1) reliance on a single primary modality for alignment, (2) pairwise modeling of modalities, and (3) assumption of complete modality observations. These limitations hinder the applicability of these approaches in real-world multimodal time-series settings, where primary modality priors are often unclear, the number of modalities can be large (making pairwise modeling impractical), and sensor failures often result in arbitrary missing observations. At its core, MAESTRO facilitates dynamic intra- and cross-modal interactions based on task relevance, and leverages symbolic tokenization and adaptive attention budgeting to construct long multimodal sequences, which are processed via sparse cross-modal attention. The resulting cross-modal tokens are routed through a sparse Mixture-of-Experts (MoE) mechanism, enabling black-box specialization under varying modality combinations. We evaluate MAESTRO against 10 baselines on four diverse datasets spanning three applications, and observe average relative improvements of 4% and 8% over the best existing multimodal and multivariate approaches, respectively, under complete observations. Under partial observations—with up to 40% of missing modalities—MAESTRO achieves an average 9% improvement. Further analysis also demonstrates the robustness and efficiency of MAESTRO's sparse, modality-aware design for learning from dynamic time series.

## 1 Introduction

Many real-world applications—such as activity recognition [2], stress monitoring [46, 39], sleep-stage detection [13], and clinical decision-making [20]—have been significantly enhanced by the integration of wearable and inconspicuous continuous-sensing technologies. Devices such as smartwatches, smartphones, and chest straps collect time-series data that encode information about various physiological and behavioral phenomena (e.g., Electrodermal Activity (EDA) for skin conductance, accelerometers for motion, and Electrocardiogram (ECG) for heart activity). Given the common temporal structure of these signals, they are often abstracted as *multivariate* time-series data [37]. However, this simplified representation can be suboptimal due to the inherent heterogeneity of sensing modalities and the varied task relevance of each modality and their interactions. An alternative is sensor fusion, but existing approaches [68, 57, 17] are often application-specific and heuristic. Even within a single application such as estimating heart rate from multimodal sensing, studies may employ diverse strategies, ranging from early-fusion in multi-wavelength [35] and multi-site PPG [34] to late fusion with temperature [36], reflecting a lack of consensus on how to systematically handle modality interactions. Hence, there is a pressing need for a general framework that can automatically learn

task-specific intra- and inter-modal dependencies without relying on ad-hoc heuristics or exhaustive fusion searches.

While significant strides have been made in the multimodal learning domain [25], most contemporary approaches that consider time-series as one of the modalities fall into one of two categories—either binding learning across multiple modalities to a single anchor modality [12, 44, 40], which naturally risks over-reliance on a predefined primary modality; or conducting explicit pairwise interaction modeling, which has shown continued promise from early efforts in cross-modal transformer, MULT [52], to the more recent FlexMoE [67]. However, such modeling becomes combinatorially expensive as the number of modalities increases—a common occurrence in rich, multi-sensor applications. Another important consideration in designing multimodal learning frameworks for time-series from real-world sensing applications is supporting learning from arbitrary combinations of sensing modalities, as sensor malfunction is commonplace [33, 47, 38].

**Our Approach.** To improve multi-modal learning for time-series data—while overcoming the limitations of existing multimodal learning frameworks and handling samples with arbitrary sensor combinations—we propose MAESTRO, which features per-Modal sparse attention with an Adaptive attention budget, followed by Efficient Sparse cross-modal attention to handle long multi-modal Temporal sequences, which are then Routed using a dynamic Mixture-of-Experts (MoE). Figure 1 presents a juxtaposition of our proposed approach and classical multivariate time-series processing in the context of a stress monitoring application, highlighting the arbitrary sensor missingness that our approach is designed

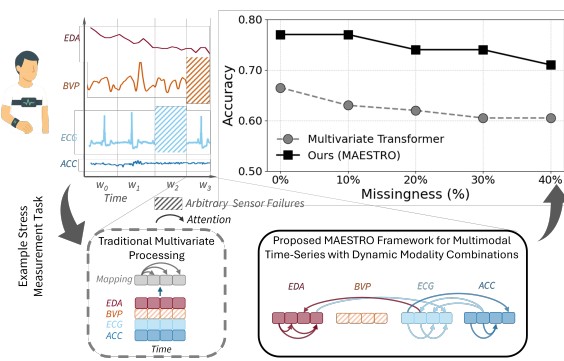

Figure 1: Illustration of traditional multivariate processing vs. our multimodal handling of sensor data, highlighting our method's superior performance and robustness.

to handle. Instead of projecting all interrelated observations at each time step into a common latent space and modeling their interactions using a classical self-attention mechanism under a multivariate setting, our MAESTRO framework applies cross-modal attention within modalities—facilitating disentangled latent projections for each modality—and explicitly models interactions across modalities at all time steps. This results in significant improvements in prediction accuracy (around 10% for the stress monitoring application; see Section 3 for detailed experimental results). More specifically, our contributions can be summarized as follows:

▸ We present a novel perspective on modeling time-series data from multiple diverse sensing modalities for multimodal learning, enabling the discovery of task-relevant, modality-specific and cross-modal interactions.

▸ We propose MAESTRO, a novel multimodal learning framework that integrates four key innovations: (1) symbolic tokenization with reserved tokens for missing data, (2) adaptive attention budgeting guided by modality availability and relevance, (3) sparse attention over long multimodal sequences to capture rich cross-modal context, and (4) loss-free, MoE-based dynamic routing that adapts to varying modality observations.

▸ We conduct extensive experiments on four real-world datasets (with 5 to 17 modalities) and demonstrate that MAESTRO consistently outperforms 10 multivariate and multimodal baselines—including state-of-the-art MoE approaches—under diverse modality availability conditions, highlighting its robust performance and its potential to advance multimodal time-series learning.

## 2   Related Works

**Multivariate and Fusion Approaches.** Approaches such as InceptionTime [16], ResNet1D [51], and Transformers [53, 10] have established themselves as leading models [37, 3] for multivariate time-series tasks. Beyond general multivariate modeling, recent research has demonstrated the benefits of variate-specific strategies—such as double attention mechanisms [29, 62, 59] and channel

selection [63, 58, 43]—which enable more granular modeling of dependencies among variables. In application-specific contexts, sensor fusion paradigms range across a spectrum, from early fusion [68]—where multivariate processing is among the earliest forms—to late fusion methods [57, 17] like ensemble learning [5], which integrate predictions at the decision level. However, these fusion strategies are often heuristic and face challenges with generalization and robustness to missing data.

**Multimodal Approach.** While multimodal learning is predominant in audio-visual domains such as speech recognition [9], video understanding [9, 1], and affective computing [24], it is increasingly being applied to fields such as robotics [19, 22, 21], human-computer interaction [8], and healthcare [11, 65, 57], which require the integration of heterogeneous time-series data from diverse sensors. Recent efforts have acknowledged this heterogeneity and proposed information-theoretic frameworks to formalize diverse multimodal interaction phenomena [54, 26, 4, 27]. Some works highlight the need for modeling both intra- and inter-modal interactions [7], while others propose explicit interaction modeling [66, 18] to capture emergent cross-modal knowledge that is task-relevant—further emphasizing the superiority of task-specific multimodal learning [70] over self-supervised multimodal representation learning [27]. Popular multimodal approaches include Tensor Fusion, multimodal Transformers, and multimodal MoEs [66, 61]. Multimodal works vary in their definition of "modality"—sometimes framing it through data encoding mechanisms [14], and other times more qualitatively, based on how the signal is manifested or experienced [4]. As a result, in prior evaluations involving time-series data, multiple sensor streams are often simplified and treated as a single modality (see Table 2 in MultiBench [25], a benchmarking resource). This treatment is suboptimal when sensors monitor fundamentally different aspects of the underlying phenomenon.

**Learning under Missing Modalities.** One of the practical challenges of multimodal learning is the unreliability of all modalities during inference. A commonly adopted approach to handle missingness is the parameterized reconstruction of missing modalities [31, 32, 50], which can be computationally intensive with increasing number of modalities. Recent multimodal approaches address this issue through similarly motivated strategies, such as missingness bank completion within the MoE framework [67, 15, 61], or cross-modal transfer—particularly when learning is achieved by binding to a primary modality [12, 40, 42] or leveraging shared modality representations [55].

## 3 Methodology

### 3.1 Preliminaries and Notations

Often time series data from distinct sources are abstracted as multivariate sequences in $\mathbb{R}^{D \times T}$, our approach explicitly models time series from different sensors as *multimodal* data. We begin by formally defining a multimodal time series:

**Definition 3.1 (Multimodal Time-series).** A multimodal time series consists of $M \geq 2$ modality-specific time series that collectively describe an underlying phenomenon through complementary, largely semantically-disjoint observations. For a sample $i \in \{1, \ldots, N\}$, we define the multimodal input as:

$$\mathcal{M}_i = \left( \underbrace{\{x_i^1[t]\}_{t=1}^{T^1}}_{\text{Modality 1}}, \ldots, \underbrace{\{x_i^M[t]\}_{t=1}^{T^M}}_{\text{Modality M}} \right),$$

where $x_i^j[t] \in \mathbb{R}^{D^j}$ denotes the feature vector at time $t$ for modality $j$, with $D^j$ features over $T^j$ time steps. Each modality $x_i^j$ forms a multivariate time series in $\mathbb{R}^{D^j \times T^j}$. The modality streams are assumed to be largely semantically-disjoint (as empirically validated later in Section 4.2), meaning that:

$$\nexists f_{jk} : \mathbb{R}^{D^j \times T^j} \to \mathbb{R}^{D^k \times T^k} \text{ such that } f_{jk}(x_i^j) \approx x_i^k, \quad \forall j \neq k.$$

**Problem Statement.** Given a multimodal time-series dataset $\mathcal{D} = \{(\mathcal{M}_i, y_i) \mid i = 1, \ldots, N\}$, where $\mathcal{M}_i = (x_i^1, \ldots, x_i^M)$ denotes a set of $M$ modality-specific time series with $x_i^j \in \mathbb{R}^{D^j \times T^j}$, and $y_i \in \{1, \ldots, C\}$ is the class label, the objective is to learn a predictor $f : \mathbb{X}_\mathcal{S} \to \{1, \ldots, C\}$, where,

$$\mathbb{X}_\mathcal{S} = \left\{ \mathcal{M}_i \mid \mathcal{M}_i = \left( \{x_i^j \mid j \in \mathcal{S}\} \right), x_i^j \in \mathbb{R}^{D^j \times T^j} \right\}$$

---

Code is available at https://github.com/payalmohapatra/MAESTRO

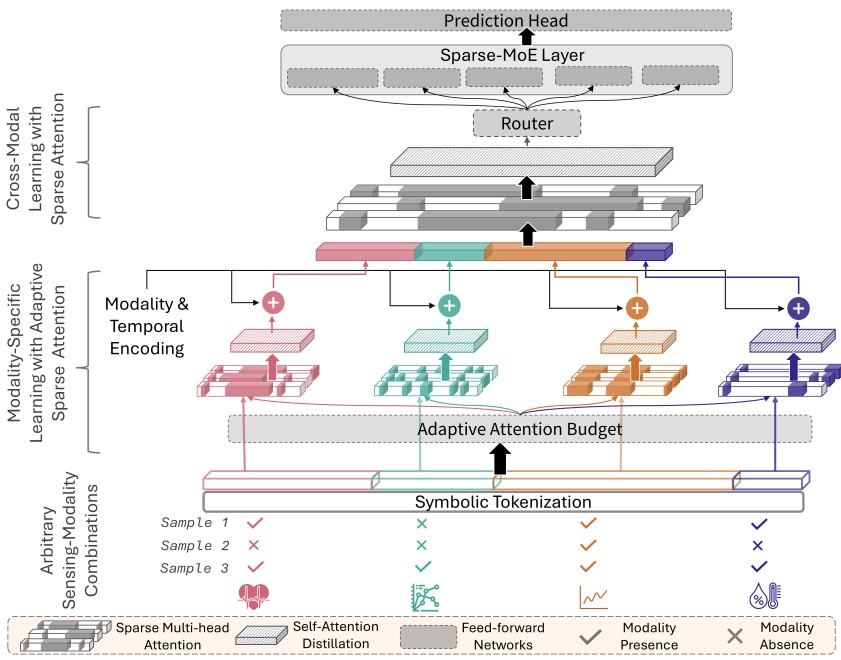

Figure 2: Overview of our approach, MAESTRO. Input data from arbitrary combinations of sensing modalities is tokenized using symbolic approximation, where a reserved symbol is used to denote missing modalities. A learnable attention budget gate to allocates modality-wise attention capacity for sparse-attention-based modality-specific encoders. The resulting modality-specific features are concatenated and combined with modality and positional embeddings, forming a long multimodal sequence, which is processed by a sparse cross-modal multihead-attention layer(s). The resulting tokens are routed through a Sparse Mixture-of-Experts module, enabling dynamic specialization under arbitrary observability conditions. Finally, a classifier maps the aggregated representation to task predictions.

for an arbitrary subset of available modalities $\mathcal{S} \subseteq \{1, \ldots, M\}$, which may vary per sample.

In this setting, not all modalities contribute equally: some are individually task-relevant, others are informative only through interactions (as characterized by Liang et al. [26]), and some may introduce noise or cause negative transfer (as empirically demonstrated later in Section 4.2). The number of possible interactions grows combinatorially with $M$, making it impractical to explicitly model all subsets in rich multisensor scenarios, where typically $M \geq 4$. Additionally, at inference time, the available modality set $\mathcal{S}_i$ may vary arbitrarily due to real-world constraints. We refer to this setting—where the combination of available sensing modalities changes across samples—as *dynamic time-series* in this paper. The objective is to learn a predictor $f$ that adapts to each $\mathcal{S}_i$, identifies and exploits informative modalities and interactions, suppresses spurious ones, and generalizes effectively across dynamic time-series.

## 3.2 Our Approach

In this section, we detail each component of MAESTRO, which is specifically designed to efficiently address key challenges in multimodal time-series learning, such as the presence of numerous sensing sources and abrupt sensor failures that lead to arbitrary combinations of available modalities, all while maintaining high performance.

We first reduce sequence length and encode missingness via symbolic representation. Next, an attention-budget gate, informed by modality relevance and availability, guides each modality-specific encoder. Their outputs are concatenated with modality and temporal position embeddings to form a unified sequence, processed by a sparse cross-modal attention network to model task-relevant interactions. The resulting tokens are routed through a Sparse MoE for final prediction. A comprehensive illustration of MAESTRO is provided in Figure 2, and the following sections present detailed design descriptions.

### 3.2.1 Missingness-aware Symbolic Tokenization for Multimodal Time Series

To represent time-series data from each modality for efficient downstream processing, we leverage the symbolic aggregate approximation (SAX) method [28]. SAX is computationally efficient and preserves pairwise relational structure. Particularly in our `MAESTRO` framework, it gives an opportunity to reserve a *symbol* to represent missingness. We first normalize the input time series $x_i^j$, and then apply piecewise aggregation to convert $x[t]$ (dropping indices $i$ and $j$ for brevity) into a compressed representation $\hat{x}[w]$:

$$\hat{x}[w] = \frac{W}{T} \sum_{t=\left(\frac{T}{W}(w-1)+1\right)}^{\frac{T}{W}w} x[t],$$

where $T$ is the length of the original time series and $W$ is the number of aggregated segments after compression.

To ensure an equiprobable distribution of symbols, we follow the empirical observation by Lin et al. [28] that normalized time series sequences follow a Gaussian distribution. We partition this distribution into $\alpha$ equal-sized areas under the Gaussian curve, referred to as regions, using breakpoints $\{\beta_0, \beta_1, \ldots, \beta_\alpha\}$, where each region corresponds to a unique symbol. The normalized piecewise aggregated value $\hat{x}[w]$ is then mapped to a symbolic token $s[w] \in \{s_1, \ldots, s_\alpha\}$ based on the breakpoint interval in which it falls, denoted using the mapping function $\phi(\hat{x}[w])$. To support arbitrary missing modalities, we reserve an additional symbol $s_0$ to indicate missingness:

$$s[w] = \begin{cases} s_0, & \text{if modality is missing at window } w \\ \phi(\hat{x}[w]), & \text{otherwise} \end{cases}.$$

Figure 3: Reconstruction from symbolic tokenization of a PPG signal.

A visualization of this transformation is shown in Figure 3. Because symbolic tokenization is applied independently to each modality, we can accommodate modality-specific sequence lengths and select compression parameters $W^j$ individually. With a shared alphabet size $\alpha$ across modalities and $s_0$ indicating missing tokens, the symbolic representation of modality $j$ for sample $i$ is:

$$s_i^j \in \{s_0, s_1, \ldots, s_\alpha\}^{D^j \times W^j},$$

where $D^j$ is the number of variates in modality $j$, and $W^j$ is the post-compression sequence length.

**Theoretical Motivation: Symbolic Representation Preserves Multimodal Relational Structure.**
We build on the `MINDIST` result from Lin et al. [28], which guarantees that the symbolic distance between time series ($\text{Dist}_{\text{sym}}$) lower-bounds their original Euclidean distance. We extend this to the multimodal setting by assuming that all modalities are normalized prior to symbolic conversion, which yields the following corollary:

**Corollary 3.2** (Cross-Modal Relational Preservation with Bounded SAX Distortion). *Let modalities $j$ and $m$ produce SAX-based symbolic representations with a shared alphabet size $\alpha$ and compression lengths $W_j$ and $W_m$, respectively. Suppose the `MINDIST` lower-bound property holds within each modality:*

$$Dist_{sym}^j(s_i^j, s_k^j) \leq \|x_i^j - x_k^j\|_2, \quad Dist_{sym}^m(s_i^m, s_k^m) \leq \|x_i^m - x_k^m\|_2,$$

*and that the SAX approximation errors are bounded:*

$$\|x_i^j - x_k^j\|_2 - Dist_{sym}^j(s_i^j, s_k^j) \leq \epsilon_j, \quad \|x_i^m - x_k^m\|_2 - Dist_{sym}^m(s_i^m, s_k^m) \leq \epsilon_m.$$

*Then, the difference in symbolic distances across modalities is bounded as:*

$$\left| Dist_{sym}^j(s_i^j, s_k^j) - Dist_{sym}^m(s_i^m, s_k^m) \right| \leq \left| \|x_i^j - x_k^j\|_2 - \|x_i^m - x_k^m\|_2 \right| + \epsilon_j + \epsilon_m.$$

**Implication.** This result above shows that SAX-based symbolic representations not only preserve relational structure within each modality, but also maintain the relative contrast in sample similarity across modalities. Specifically, if two samples are more similar in modality $j$ than in modality $m$, this relationship is preserved after symbolic discretization. This implies that symbolic tokenization retains meaningful cross-modal structure. A proof sketch is provided in Appendix A.

### 3.2.2 Adaptive Sparse Attention for Intra-modal Learning

Following the tokenized representations described in Section 3.2.1, we extract task-specific intra-modal features using dedicated encoders for each modality. Based on the standard formulation of Vaswani et al. [53], we first apply sinusoidal positional embeddings ($\text{PE}_{\text{sin}}$) to the modality stream $s^j$ and compute the query, key, and value matrices $\mathbf{Q}^j \in \mathbb{R}^{d \times L_{Q^j}}$, $\mathbf{K}^j \in \mathbb{R}^{d \times L_{K^j}}$, $\mathbf{V}^j \in \mathbb{R}^{d \times L_{V^j}}$. However, instead of the canonical self-attention mechanism, we employ a sparse self-attention strategy, $\mathcal{A}_s(\mathbf{Q}^j, \mathbf{K}^j, \mathbf{V}^j)$, inspired by Zhou et al. [69], with a key modification: the sparsity budget is adaptively controlled by the learned function $a(\mathbf{m}_i; \theta_a)$, where $\mathbf{m}_i \in \mathbb{R}^{1 \times M}$ is a logit vector indicating the presence of each modality for an input sample $s_i^j$. Details on training $a(\mathbf{m}_i; \theta_a)$ are provided in Section 3.2.3. We denote the budget per modality as $u = \mathbf{u}_i[j]$. For brevity, we omit indices $i$ and $j$ in the following description of modality-specific encoders with adaptive sparse attention.

For each query $\mathbf{q} \in \mathbf{Q}$, a random subset of keys denoted as $\mathbf{K}' \subset \mathbf{K}$, is sampled where $|\mathbf{K}'| = u \log L_K$ [69]. The dot product $\mathbf{q} \cdot \mathbf{K}'$ is used to compute the max-mean sparsity metric $\mathcal{P}(\mathbf{q}, \mathbf{K}')$ proposed by Zhou et al. [69] given below :

$$P(\mathbf{q}, K') = \max_{\mathbf{k} \in K'} \left( \frac{\mathbf{q}\mathbf{k}^\top}{\sqrt{d}} \right) - \frac{1}{|K'|} \sum_{\mathbf{k} \in K'} \frac{\mathbf{q}\mathbf{k}^\top}{\sqrt{d}}. \tag{1}$$

This max-mean measurement evaluates the query's attention diversity. The queries with higher $P(\mathbf{q}, K')$ scores contain more distinctive information. Next, the top-$\upsilon$ queries—where $\upsilon = u \log L_Q$—with the highest sparsity scores are selected to favor more *diverse* queries [69]. The top-$\upsilon$ queries are selected independently for each attention head to avoid excessive information loss due to sparsification. We then `distil` the self-attention outputs using 1D convolutional layers and max-pooling followed by residual connections. The overall structure of the modality-specific encoders, $g \colon (s, \mathbf{u}) \mapsto z$, is summarized below:

$$
\begin{aligned}
\hat{s} &= s + \text{PE}_{\text{sin}}(s) && \text{(positional encoding)} \\
\bar{s} &= \mathcal{A}_s(\mathbf{Q}, \mathbf{K}, \mathbf{V}) = \texttt{Softmax}\left( \frac{\bar{\mathbf{Q}}\mathbf{K}^\top}{\sqrt{d}} \right)\mathbf{V} && \text{(sparse multi-head attention; } \bar{\mathbf{Q}} \text{ is sparse)} \\
\dot{s} &= \bar{s} + \hat{s} && \text{(residual connection)} \\
z &= \texttt{distil}(\dot{s}) + \texttt{maxpool}(\hat{s}) && \text{(attention distillation)}
\end{aligned}
\tag{2}
$$

### 3.2.3 Modality-Aware Attention Budgeting

Our modality-specific encoders previously described are designed using a sparse attention mechanism. We propose to adaptively learn this sparse attention budget for each modality based on its task relevance and availability by modulating the maximum attention budget, denoted by $\beta \in \mathbb{R}^+$, resulting in a budget vector $\mathbf{u}_i \in \mathbb{R}^{1 \times M}$. The attention budget $\mathbf{u}_i$, parameterized by $\theta_a$, is computed as:
$$\mathbf{u}_i = \lfloor \sigma\left(\mathcal{G}(\mathbf{m}_i + \epsilon(1 - \mathbf{m}_i); \theta_a)\right) \cdot \beta \rfloor,$$
where $\epsilon$ is a small constant used for numerical stability, and $\sigma(\cdot)$ denotes the sigmoid activation. The gating function $\mathcal{G}(\mathbf{m}_i; \theta_a)$ adaptively allocates attention capacity based on both modality availability and task relevance, as it is trained with the task-specific objective.

### 3.2.4 Cross-Modal Learning using Sparse Attention for Long Multimodal Sequences

Let $\{\mathbf{z}_j\}_{j=1}^M$ denote the modality-specific embeddings obtained from the previously described encoders. At this stage, our goal is to devise an optimal learning strategy for inter-modal interactions that are relevant to the task while avoiding the effect of noisy or task-irrelevant modalities that may degrade predictions (illustrated with a case-study in Section 4.2).

**Constructing the Multimodal Sequence.** We concatenate features along the temporal dimension to form a unified representation. Following standard practice [64], we first add modality-specific and temporal positional embeddings to each $\mathbf{z}_j$, denoted as $\mathbf{ME}(\mathbf{z}_j)$ and $\mathbf{PE}(\mathbf{z}_j)$, respectively: $\hat{\mathbf{z}}_j = \mathbf{z}_j + \mathbf{ME}(\mathbf{z}_j) + \mathbf{PE}(\mathbf{z}_j)$. These representations are then concatenated along the temporal axis to form a unified multimodal sequence $\mathbf{c} = \texttt{Concat}_{\text{time}}(\hat{\mathbf{z}}_1, \hat{\mathbf{z}}_2, \ldots, \hat{\mathbf{z}}_M)$, where $M$ is the number of modalities, and $\mathbf{c} \in \mathbb{R}^{\hat{D} \times \hat{L}}$, with $\hat{L} = \sum_{j=1}^M L_j$.

This strategy offers the some key advantages: (1) it supports varying sequence lengths for different modalities. For example, in activity recognition tasks, accelerometers typically have a higher sampling rate (generally 25 Hz) than electrodermal activity sensors (generally 4 Hz). Uniformly resampling them can be suboptimal; (2) it facilitates time-varying cross-attention—queries from one modality can attend to keys from the same modality as well as from other modalities. This acts as a generalized form of self-attention within a modality and cross-attention across modalities, enabling the learning of relevant inter-modal interactions; and (3) it is independent of explicit pairwise interaction modeling, which is impractical for modalities $\geq 4$, as is typically the case in sensing applications. However, one of the key challenges with this approach is the increased sequence length, where $\hat{L} = \sum_{j=1}^{M} L_j$, especially as the number of modalities grows. To address this, we propose leveraging the sparse-attention mechanism to handle this long multimodal sequence.

**Applying Sparse-Cross-Attention.** To capture inter-modal dependencies, we apply sparse attention to the concatenated sequence $\mathbf{c}$. Let $\bar{\mathbf{c}} = \mathcal{A}_s(Q_c, K_c, V_c)$, where $\mathcal{A}_s$ is the sparse multi-head attention operator, and $Q_c$, $K_c$, and $V_c$ are the query, key, and value vectors derived from $\mathbf{c}$. The sparsity budget is controlled by a fixed parameter $\beta$ (similar to $u = 1$, in the context of the modality-specific sparse-attention encoders in Section 3.2.2). This sparse attention mechanism reduces the computational complexity from $\mathcal{O}(\hat{L}^2)$ to $\mathcal{O}(\hat{L} \log \hat{L})$, adeptly handling this long concatenated multimodal sequence. Table 1 summarizes

Table 1: Cross-attention methods' complexity. $\hat{L}$: multimodal sequence length, $M$: number of modalities, $L_{\max}$: longest modality sequence length.

| Method | Time | Space |
|---|---|---|
| Dense | $\mathcal{O}(\hat{L}^2)$ | $\mathcal{O}(\hat{L}^2)$ |
| Pairwise | $\mathcal{O}(M^2 L_{\max}^2)$ | $\mathcal{O}(M^2 L_{\max}^2)$ |
| Sparse (Ours) | $\mathcal{O}(\hat{L} \log \hat{L})$ | $\mathcal{O}(\hat{L} \log \hat{L})$ |

the space-time complexity of our proposed cross-modal sparse attention-based framework against its dense and pairwise cross-modal counterparts. After applying sparse attention, $\bar{\mathbf{c}}$ undergoes similar `distil` transformation and residual connections, described in Equation (2), yielding the final output $\mathbf{e} \in \mathbb{R}^{\hat{D} \times \hat{L}}$. Figure 9 in Section 4.2 illustrates an instance of $\mathbf{e}$ and confirms its utility in capturing cross-modal relations.

### 3.2.5 Sparse Mixture-of-Experts Routing and Optimization

Inspired by the success of curriculum learning strategies in recent works on earning from arbitrary multimodal input data [67], we initially train the model on modality-complete samples and gradually expose it to more challenging examples with missing modalities. Specifically, the modality dropout probability $p(\tau) = \min\left(p_{\max}, \frac{\tau - \tau_{\text{warmup}}}{\tau_{\max} - \tau_{\text{warmup}}} \cdot p_{\max}\right)$ increases linearly with the epoch index $\tau$ after a warm-up period $\tau_{\text{warmup}}$, and is capped at a maximum value $p_{\max}$.

To introduce input-dependent dynamism based on modality combinations, we adopt the standard Sparse MoE [48] layer to process the input representation $e$. The MoE layer consists of $\Omega$ *experts*—fully connected layers—and a router trainable $\mathcal{R}$, which selects the top-$k$ experts to process each of the $\hat{L}$ tokens. The progressive modality-dropout acts as a form of regularization [60], encouraging implicit expert specialization without requiring any explicit auxiliary load-balancing losses. Such loss-free MoE optimization is also supported by recent works [23, 6, 56]. Our empirical observations (see Figure 4) suggest this specialization behavior: in a setting with $\Omega = 8$ experts and $k = 1$, the routing patterns of $\mathcal{R}$ exhibit variability across different combinations of input modalities at inference

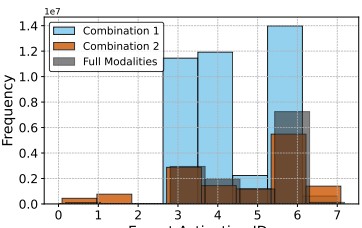

Figure 4: Expert routing decisions across different input modality combinations, highlighting the implicit specialization behavior of the sparse MoE-layer.

time. These results align with our design intent (as verified in Figure 6 in Section 4.2), indicating that MoEs can enable black-box specialization and dynamic routing based on input conditions. We use the final logit predictions from the selected experts (typically with $k = 1$; for $k > 1$, we aggregate logits via averaging) to compute the task loss. Given aggregated logits $\hat{y} \in \mathbb{R}^C$, we first apply softmax normalization to obtain class probabilities $p_c = \frac{\exp(\hat{y}_c)}{\sum_{i=1}^{C} \exp(\hat{y}_i)}$. For ground-truth label $y \in \{1, \dots, C\}$, the cross-entropy loss is: $\mathcal{L}_{\text{CE}} = -\log p_y$. Detailed implementation and optimization settings are described in Appendix C.

## 4 Results and Discussion

We evaluate MAESTRO against **10 state-of-the-art methods on four multimodal time-series datasets across three application domains**, using per-segment accuracy and macro-F1 score.

**Datasets.** We leverage four multimodal time-series datasets spanning diverse sensing configurations and tasks. **WESAD** [46] includes 10 chest- and wrist-mounted modalities for 3-class cognitive stress classification. **DaliaHAR**, constructed from the Dalia dataset [45], is a 7-class physical activity recognition task using five physiological and motion modalities. **DSADS** [2] comprises 9-axis IMU signals from five body locations for 19-class activity classification. **MIMIC-III**, processed via MultiBench [25], is a clinical dataset for 6-class diagnostic prediction using 17 modalities. Additional dataset statistics and preprocessing steps are detailed in Appendix D.2.

**Baselines.** We compare MAESTRO against strong baselines spanning multivariate, multimodal, and missingness-aware approaches. From recent benchmark leaders [37], we include **InceptionTime** [16], **ResNet1D** [10], and **Transformer** [53], along with **iTransformer** [29], which leverages inverted attention. For multimodal settings, we evaluate late fusion approaches including **ensemble learning** [5] and **low-rank tensor fusion** (LRTF) [30]. We further include MoE-based frameworks that support missing modalities—**FlexMoE** [67] and **FuseMoE** [15]—as well as interaction-centric models like **MULT** [52] and **ShaSpec** [55], motivated by [26, 25]. To evaluate robustness under modality dropout, we apply the missingness-aware scheme from Section 3.2.5 to Transformer, and compare against natively-missingness resilient baselines—FlexMoE, FuseMoE, and ShaSpec—under varying amounts of missingness (10% to 40%). All baselines use published hyperparameters or undergo search when unspecified. Each dataset has three distinct splits (80% train, 10% valid and 10% test), with three trials per split. We report mean performance in the main text; full statistics and implementation details are given in Appendix E and D, respectively.

### 4.1 Advantages of MAESTRO: Superior Performance and Robustness to Arbitrary Missingness

Table 2 reports the performance of MAESTRO against multivariate and multimodal baselines under full modality availability, with the following key observations: (1) MAESTRO achieves consistent gains over the best baselines—8% relative improvement on WESAD, 5% on DaliaHAR, and 4% on DSADS; (2) the MIMIC dataset yields lower absolute performance and smaller gains due to its task complexity—which aligns with previously reported findings [25]. To better interpret MIMIC results, we analyze macro-F1 scores: MAESTRO achieves 0.30, outperforming the generally strong baselines FuseMoE and FlexMoE (both at 0.27), reflecting an 11% relative improvement; (3) using symbolic representation of the multimodal input preserves semantic structure and improves the overall performance of the MAESTRO framework by 6% relatively under full modality observability; (4) overall, multimodal approaches outperform multivariate ones by an average relative improvement of 4% across all datasets. MAESTRO **delivers consistent relative improvements—8% over top multivariate and 4% over top multimodal baselines—across all benchmarks.**

Table 2: Performance (Accuracy/F1-score) comparison across datasets under full modality observability. Best per dataset in **bold and orange**, second-best in *italics and underlined*. Mean$_{\pm}$std reported.

| Model Type | Model | WESAD | | DaliaHAR | | DSADS | | MIMIC III | |
|---|---|---|---|---|---|---|---|---|---|
| | | Acc ↑ | F1 ↑ | Acc ↑ | F1 ↑ | Acc ↑ | F1 ↑ | Acc ↑ | F1 ↑ |
| **Multivariate** | InceptionTime | $0.59_{\pm0.02}$ | $0.51_{\pm0.03}$ | $0.73_{\pm0.03}$ | $0.68_{\pm0.04}$ | $0.81_{\pm0.02}$ | $0.81_{\pm0.02}$ | $0.74_{\pm0.01}$ | *$0.29_{\pm0.01}$* |
| | Transformer | $0.63_{\pm0.01}$ | $0.53_{\pm0.02}$ | $0.76_{\pm0.02}$ | $0.71_{\pm0.03}$ | $0.83_{\pm0.01}$ | $0.83_{\pm0.01}$ | *$0.78_{\pm0.02}$* | $0.22_{\pm0.01}$ |
| **Cross-variate** | ResNet1D | $0.52_{\pm0.03}$ | $0.44_{\pm0.03}$ | $0.73_{\pm0.03}$ | $0.69_{\pm0.03}$ | $0.79_{\pm0.02}$ | $0.78_{\pm0.02}$ | $0.77_{\pm0.02}$ | $0.18_{\pm0.01}$ |
| | iTransformer | $0.67_{\pm0.02}$ | $0.53_{\pm0.02}$ | $0.69_{\pm0.02}$ | $0.66_{\pm0.02}$ | $0.62_{\pm0.03}$ | $0.61_{\pm0.03}$ | $0.77_{\pm0.01}$ | $0.14_{\pm0.01}$ |
| **Multimodal(MM)** | LRTF | $0.52_{\pm0.01}$ | $0.30_{\pm0.06}$ | $0.48_{\pm0.04}$ | $0.19_{\pm0.03}$ | $0.72_{\pm0.09}$ | $0.70_{\pm0.12}$ | $0.77_{\pm0.01}$ | $0.15_{\pm0.01}$ |
| | Ensemble | $0.69_{\pm0.11}$ | $0.57_{\pm0.01}$ | $0.74_{\pm0.12}$ | $0.71_{\pm0.11}$ | $0.59_{\pm0.07}$ | $0.57_{\pm0.11}$ | *$0.78_{\pm0.04}$* | $0.27_{\pm0.05}$ |
| **MM MoE** | FuseMoE | $0.47_{\pm0.03}$ | $0.41_{\pm0.03}$ | *$0.79_{\pm0.02}$* | *$0.79_{\pm0.02}$* | *$0.85_{\pm0.01}$* | *$0.85_{\pm0.01}$* | $0.75_{\pm0.02}$ | $0.27_{\pm0.01}$ |
| | FlexMoE | *$0.71_{\pm0.13}$* | *$0.63_{\pm0.09}$* | $0.70_{\pm0.06}$ | $0.70_{\pm0.05}$ | $0.70_{\pm0.06}$ | $0.68_{\pm0.09}$ | **$0.79_{\pm0.03}$** | $0.27_{\pm0.03}$ |
| **MM Interaction** | MULT | $0.60_{\pm0.03}$ | $0.42_{\pm0.02}$ | $0.72_{\pm0.02}$ | $0.72_{\pm0.02}$ | $0.66_{\pm0.04}$ | $0.65_{\pm0.03}$ | $0.79_{\pm0.01}$ | $0.21_{\pm0.01}$ |
| | ShaSpec | $0.62_{\pm0.03}$ | $0.51_{\pm0.02}$ | $0.75_{\pm0.02}$ | $0.78_{\pm0.02}$ | $0.82_{\pm0.01}$ | $0.81_{\pm0.01}$ | $0.74_{\pm0.01}$ | $0.24_{\pm0.01}$ |
| **Ours** | without SAX | $0.69_{\pm0.02}$ | $0.55_{\pm0.02}$ | *$0.82_{\pm0.01}$* | **$0.84_{\pm0.01}$** | $0.78_{\pm0.02}$ | $0.77_{\pm0.02}$ | **$0.79_{\pm0.01}$** | **$0.30_{\pm0.01}$** |
| | with SAX | **$0.77_{\pm0.02}$** | **$0.66_{\pm0.01}$** | **$0.83_{\pm0.01}$** | **$0.84_{\pm0.01}$** | **$0.88_{\pm0.01}$** | **$0.88_{\pm0.01}$** | *$0.78_{\pm0.01}$* | **$0.30_{\pm0.01}$** |

MAESTRO is pragmatically designed to handle arbitrary modality missingness through reserved symbolic representations, adaptive attention budgeting, cross-modal learning, and modality dropout

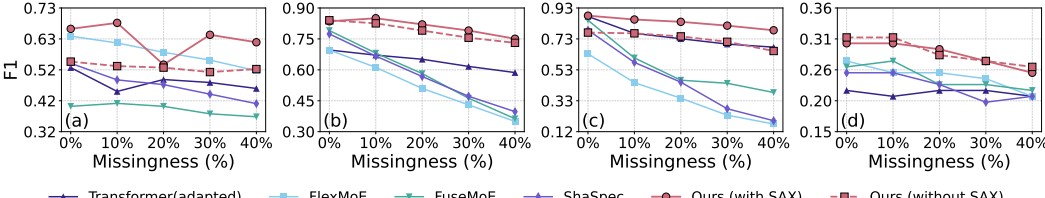

Figure 5: Comparative performance (Macro-F1 score) of `MAESTRO` against missingness-aware multimodal baselines and a modality-dropout-adapted Transformer shows its consistent superiority across varying missingness levels on (a) WESAD, (b) DaliaHAR, (c) DSADS, and (d) MIMIC-III.

training. As shown in Figure 5: (1) under 40% missingness, `MAESTRO` outperforms the strongest missingness-resilient multimodal baseline by an average relative F1 gain of 59% across datasets; (2) a representative multivariate model—Transformer adapted with the same missingness-aware training strategy as `MAESTRO`—generally outperforms other multimodal baselines, yet `MAESTRO` still exceeds its performance by 25% relatively; and (3) symbolic reservation for missing modalities in `MAESTRO` yields an average relative gain of 11%. **Across all datasets, `MAESTRO` achieves increasing accuracy improvements over the best baseline—7.6% absolute improvement under 10% missingness and 9.4% under 40%**. Detailed results are in Appendix E.

## 4.2 Architecture, Efficiency, and Empirical Case Study of `MAESTRO`

We further analyze `MAESTRO` through ablation and complexity studies on the WESAD dataset.

**Ablation Study.** We examine the key components of `MAESTRO` that drive its performance under both full and partial (40% missingness) modality settings. As shown in Figure 6, symbolic tokenization, modality-specific positional embeddings, and sparse MoE routing each contribute substantial gains—ranging from 5% to 22%. While modality dropout has limited impact under full modality (2%), it is critical under missingness, where its removal causes a 9% drop. The 22% drop without the sparse MoE under incomplete observations further supports our design intuition that the router ($\mathcal{R}$) implicitly specializes for different modality combinations, as illustrated in Figure 4 (Section 3.2.5).

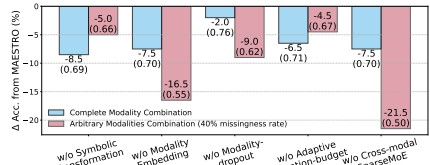

Figure 6: Ablation of `MAESTRO`: absolute accuracy drop ($\Delta$) from full model, with ablated scores in parentheses.

**Complexity Study.** To demonstrate the efficiency of `MAESTRO`, we compute giga floating-point operations (GFLOPs), multiply-accumulate operations (MMACs) and number of trainable parameters under the complete-modality setting. From Table 3, we observe: (1) replacing sparse with full attention in `MAESTRO`'s components yields at most an absolute performance improvement of 3%, while reducing GFLOPs by an average of 20%; (2) multivariate handling is slightly more efficient (0.4 GFLOPs lower) but suffers from nearly 10% lower performance; and (3) `MAESTRO` outperforms existing multimodal frameworks in both accuracy and efficiency for multimodal time-series. In particular, juxtaposing it against MULT underscores the cost of pairwise-exhaustive modeling, which consumes nearly 200% more GFLOPs than our proposed `MAESTRO`.

Table 3: Computational Complexity. In `MAESTRO`, sparse attention in the Per-Modal and Cross-Modal components is replaced by dense attention, referred to as Full-Attn (Per-Modal) and Full-Attn (Cross-Modal), respectively. Replacing all sparse attention components with dense attention is denoted as All Full-Attn.

| Model | Acc. ↑ | MMAC ↓ | GFLOPs ↓ | Params (M) |
|---|---|---|---|---|
| *Multivariate Models* | | | | |
| iTransformer | $0.67_{\pm 0.05}$ | 2833 | 5.73 | 12.82 |
| Transformer | $0.63_{\pm 0.02}$ | 4331 | 8.66 | 1.68 |
| *Multimodal Models* | | | | |
| FuseMoE | $0.47_{\pm 0.41}$ | 6524 | 13.05 | 0.67 |
| MULT | $0.60_{\pm 0.42}$ | 13324 | 26.65 | 3.71 |
| ShaSpec | $0.62_{\pm 0.51}$ | 4556 | 9.11 | 216 |
| **MAESTRO** | $0.77_{\pm 0.0c}$ | 3066 | 6.13 | 1.39 |
| – Full-Attn (Per-Modal) | $0.80_{\pm 0.03}$ | 3769 | 7.54 | 1.40 |
| – Full-Attn (Cross-Modal) | $0.77_{\pm 0.07}$ | 3496 | 6.99 | 1.39 |
| – All Full-Attention | $0.75_{\pm 0.05}$ | 4205 | 8.42 | 1.39 |
| – All Full-Attention (no MoE) | $0.78_{\pm 0.04}$ | 4392 | 8.78 | 1.39 |

**Sensitivity Study.** We present sensitivity studies for the model parameters—compression ratio ($\frac{T}{W}$) and expert count ($\Omega$)—in Figure 7 (with details in Tables 16 and 17 in the Appendix), as well as for input noise. For input noise, we design two pilot experiments. In the first, shown in Figure 8, we show that the symbolic transformation not only enables input compression and symbol reservation (Section 3.2.1), but also mitigates small local perturbations such as Gaussian noise. Next, in a controlled study using simple additive noise (described in detail in Section E.3) in fixed modalities in Table 4, we observe that even for aggressive noise such as simulated electrical interference spikes, `MAESTRO` shows similar performance to full-modality scenario in one combination and in another its close to when the modalities are completely missing. This supports our multimodal treatment of time-series from heterogeneous sensors, with a cross-attention-based design and Missingness-aware

Symbolic Tokenization that enables querying only the relevant inter- and intra-modal values for a given task. This provides inherent robustness to missing modalities, and through these controlled experiments, we can also observe `MAESTRO`'s robustness to other simple perturbations. We also include a preliminary time-dependent noise analysis (Section E.4 of the Appendix) to demonstrate `MAESTRO`'s ability to handle asynchronous inputs, and a high-density sampling study (Section E.5 of the Appendix) to highlight its computational benefits for long-range multimodal time series.

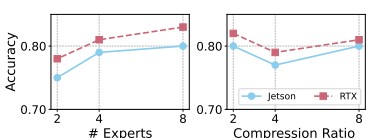 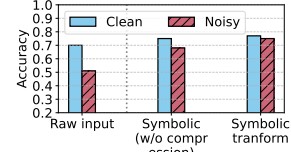

Figure 7: Sensitivity Analysis.

Figure 8: Comparing the differen input representations with noisy input.

Table 4: Performance under simple noise. Comb. 1 omits `chest_acc` and Comb. 2 omits `wrist BVP, EDA, and Temp.`

| Corruption | Comb. 1 | Comb. 2 |
|---|---|---|
| *None* | 0.77 | 0.77 |
| Missing | 0.61 | 0.65 |
| Random Noise | 0.61 | 0.63 |
| Add. Noise ($\sigma = 0.5$) | 0.75 | 0.74 |
| Add. Noise + Spikes | 0.60 | 0.73 |

Several datasets exhibit large performance differences between the best and worst unimodal models (e.g., 35% in WESAD, 59% in DaliaHAR). In WESAD (a 3-class task), the temperature modality performs near random, with an accuracy of 0.38. Full unimodal results are reported in Appendix F.1. Some datasets, like DSADS, exhibit higher redundancy, while in MIMIC, modality gaps are minimal, though overall accuracy remains low. These trends support our design intuition: *modality heterogeneity is inherent*. `MAESTRO` addresses this diversity and remains competitive with the best unimodal models, offering average relative gains of 7%.

**Case study on DaliaHAR.** For activity recognition, motion sensors (`wrist_ACC`, `chest_ACC`) are clearly more relevant than other physiological sensors. We perform a combinatorially exhaustive study ($\sum \binom{M}{i}$) and observe that: (1) some modalities—`wrist_BVP` and `wrist_Temp`—frequently appear in the lower-performance category (accuracy $\leq 0.70$) (Table 5), and the averaged cross-modal attention map, $\mathbf{e}$, from Section 3.2.4 uncovers such relational patterns (Figure 9); (2) *a priori* understanding of the most informative modalities further improves `MAESTRO`'s performance from 0.83 to 0.85 average accuracy—highlighting that an *a priori*-guided `MAESTRO` can enhance predictive performance without any additional updates (complete results in Appendix F).

Table 5: Low ($\leq 0.70$) vs High ($>0.77$) performance combinations for DaliaHAR.

| Low-performing Combinations | Acc. | High-performing Combinations | Acc. |
|---|---|---|---|
| chest_ACC, wrist_EDA, wrist_TEMP | 0.66 | chest_ACC, wrist_ACC | 0.80 |
| wrist_ACC, wrist_BVP, wrist_TEMP | 0.67 | wrist_ACC | 0.79 |
| chest_ACC, wrist_BVP, wrist_TEMP | 0.68 | chest_ACC, wrist_EDA | 0.79 |
| chest_ACC, wrist_BVP | 0.67 | chest_ACC, wrist_ACC, wrist_EDA | 0.79 |
| wrist_ACC, wrist_BVP | 0.64 | wrist_ACC, wrist_EDA | 0.77 |
| chest_ACC, wrist_ACC, wrist_TEMP | 0.70 | chest_ACC | 0.85 |

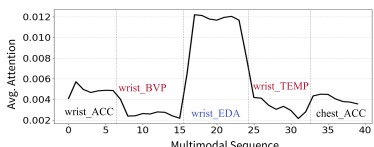

Figure 9: Averaged Attention map for a batch across the multimodal sequence.

## 5 Conclusion

We propose a novel multimodal framework, `MAESTRO`, to model time-series from heterogeneous sensing modalities and address the limitations of existing multivariate approaches—such as oversimplified feature interactions—and existing multimodal approaches, which often rely on pairwise interactions or overemphasize a single dominant modality. `MAESTRO` achieves this by using symbolic representations to tokenize both time-series data and missingness, applying adaptive budgeted intra-modal learning based on modality availability and relevance, and leveraging sparse cross-modal attention to capture multimodal interactions. `MAESTRO` is specifically designed to handle input dynamism, including arbitrary modality missingness. Extensive evaluations demonstrate `MAESTRO`'s superior performance, robustness, and overall efficiency.

**Broader Impact and Future Work.** `MAESTRO` paves the way for more efficient and pragmatic handling of heterogenous sensing data. Currently, we handle complete missingness in modalities. In future, we aim to explore more advanced symbolic encoding strategies and extend the framework to address irregularly sampled and asynchronous sensing modalities.

## 6 Acknowledgment

We gratefully acknowledge support in part from the National Science Foundation under Grants 2038853, 2324936, and 2328973.

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

# Appendix

This Appendix includes additional details for the paper, "`MAESTRO` : Adaptive Sparse Attention and Robust Learning for Multimodal Dynamic Time Series", including the reproducibility statement, additional details on symbolic tokenization and theoretical proof of **Corollary 3.2** of the main paper (Section A), additional details for sparse multihead attention (Section B), additional experimental setup and training details of `MAESTRO` (Sections C and D) with detailed dataset introduction (Section D.2), more detailed results of the experiments shown in the main paper (Section E), additional experiments (Section F), and more discussion on broader impacts.

## Reproducibility Statement

A minimal source-code has been provided in the Supplementary Materials. We use public datasets and provide implementation details in the following sections.

## A    Additional Details on the Symbolic Tokenization

Piecewise Aggregate Approximation (PAA) and Symbolic Aggregate approXimation (SAX) are sequential time-series compression methods that reduce temporal resolution by dividing the signal into fixed-size windows. PAA summarizes each window by its mean value, producing a smoothed, real-valued lower-dimensional representation. SAX builds on PAA by further discretizing these means into symbolic tokens using breakpoints derived from a standard Gaussian distribution, resulting in a compact and interpretable symbolic sequence. As detailed in Section 3.2.1 of the main paper, PAA serves as an intermediate step in the SAX transformation pipeline, bridging raw signal compression and symbolic quantization. While PAA captures coarse temporal structure, SAX enables symbolic reasoning, efficient indexing, and explicit missingness encoding via a reserved symbol. An illustration for their comparison is shown in Figure 10.

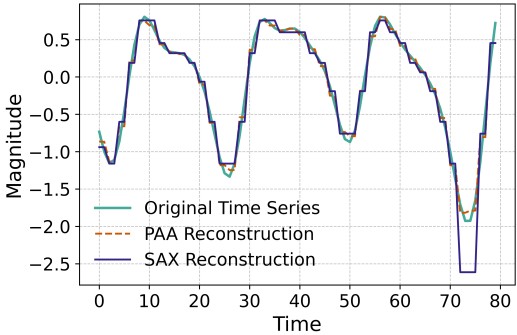

Figure 10: Comparison of PAA and SAX representations. The raw photoplethysmograph (PPG) signal $x[t]$ is first segmented and averaged into $\bar{x}[w]$ via PAA. SAX extends this by mapping each PAA segment to a discrete symbol $s[w] \in \{s_1, \ldots, s_\alpha\}$, yielding a compact symbolic sequence. A reserved token $s_0$ can be used to indicate missing segments.

**Proof Sketch for Corollary in Section 3.2.1 (Cross-Modal Relational Preservation)**

**Goal.** Show that:
$$\left| \text{Dist}_{\text{sym}}^j(s_i^j, s_k^j) - \text{Dist}_{\text{sym}}^m(s_i^m, s_k^m) \right| \leq \left| \|x_i^j - x_k^j\|_2 - \|x_i^m - x_k^m\|_2 \right| + \epsilon_j + \epsilon_m.$$

**Assumptions.**

1. **Lower-bound property of SAX (MINDIST):**
$$\text{Dist}_{\text{sym}}^j(s_i^j, s_k^j) \leq \|x_i^j - x_k^j\|_2, \quad \text{Dist}_{\text{sym}}^m(s_i^m, s_k^m) \leq \|x_i^m - x_k^m\|_2.$$

2. **Bounded symbolic approximation error:**
$$\|x_i^j - x_k^j\|_2 - \text{Dist}_{\text{sym}}^j(s_i^j, s_k^j) \leq \epsilon_j, \quad \|x_i^m - x_k^m\|_2 - \text{Dist}_{\text{sym}}^m(s_i^m, s_k^m) \leq \epsilon_m.$$

**Proof.** We begin by applying the *triangle inequality*, which states that for any real numbers $a, b, c$, the following holds:
$$|a - c| \leq |a - b| + |b - c|.$$
We use this property iteratively to decompose the difference between the symbolic distances.

$$\left|\text{Dist}_{\text{sym}}^j - \text{Dist}_{\text{sym}}^m\right| = \left|\text{Dist}_{\text{sym}}^j - \|x_i^j - x_k^j\|_2 + \|x_i^j - x_k^j\|_2 - \|x_i^m - x_k^m\|_2 + \|x_i^m - x_k^m\|_2 - \text{Dist}_{\text{sym}}^m\right|$$

$$\leq \left|\text{Dist}_{\text{sym}}^j - \|x_i^j - x_k^j\|_2\right| + \left|\|x_i^j - x_k^j\|_2 - \|x_i^m - x_k^m\|_2\right| + \left|\|x_i^m - x_k^m\|_2 - \text{Dist}_{\text{sym}}^m\right|.$$

By the assumption of bounded symbolic error, we have:
$$\left|\text{Dist}_{\text{sym}}^j - \|x_i^j - x_k^j\|_2\right| \leq \epsilon_j, \quad \left|\text{Dist}_{\text{sym}}^m - \|x_i^m - x_k^m\|_2\right| \leq \epsilon_m.$$

Substituting these bounds, we obtain:
$$\left|\text{Dist}_{\text{sym}}^j - \text{Dist}_{\text{sym}}^m\right| \leq \left|\|x_i^j - x_k^j\|_2 - \|x_i^m - x_k^m\|_2\right| + \epsilon_j + \epsilon_m.$$

## B   Additional Details for Sparse Multihead Attention

We adopt the sparsity measurement proposed by Zhou et al. [69] to efficiently identify dominant queries without computing all query-key pairs. For each query vector $\mathbf{q} \in Q$, we compute its sparsity score $P(\mathbf{q}, K')$ over a sampled subset of keys $K' \subset K$ where $|K'| = u \log L_K$:

$$P(\mathbf{q}, K') = \max_{\mathbf{k} \in K'} \left(\frac{\mathbf{q}\mathbf{k}^\top}{\sqrt{d}}\right) - \frac{1}{|K'|} \sum_{\mathbf{k} \in K'} \frac{\mathbf{q}\mathbf{k}^\top}{\sqrt{d}}. \tag{3}$$

This max-mean measurement evaluates the query's attention diversity by comparing its maximum alignment with the average alignment over the key subset. Queries with higher $P(\mathbf{q}, K')$ scores contain more distinctive information and are prioritized in our sparse attention mechanism.

The top-$v$ queries with $v = u \log L_Q$ highest scores are selected for full attention computation, reducing the complexity from $O(L_Q L_K)$ to $O(u \log L_Q \cdot u \log L_K)$. This adaptive selection enables efficient processing while preserving the most informative query-key interactions.

## C   Training and Optimization Details

All experiments are performed on an Ubuntu OS server equipped with NVIDIA TITAN RTX GPU cards using PyTorch framework. Every experiment is carried out with 3 different seeds (2711, 2712, 2713). During model training, we use Adam optimizer with a learning rate from 1e-5 to 1e-3 and maximum number of epochs is set to 150 based on the suitability of each setting. We tune these optimization-related hyperparameters for each setting and save the best model checkpoint based on early exit based on the minimum value of the loss function achieved on the validation set.

**Modality Dropout Scheme based on Curriculum Learning.** As described in Section 3.2.5, the modality dropout is illustrated in Figure 11.

**Hyperparameters.** The key hyperparameters in `MAESTRO` are listed in Table 6 and are kept fixed across all datasets in this paper. However, a more comprehensive hyperparameter search could potentially yield further improvements in performance.

## D   Experimental Setup Details

### D.1   Performance Metrics

**Accuracy.**   Given $N$ samples, Accuracy is defined as the proportion of correct predictions:

$$\text{Accuracy} = \frac{1}{N} \sum_{i=1}^{N} \mathbb{K}(\hat{y}_i = y_i)$$

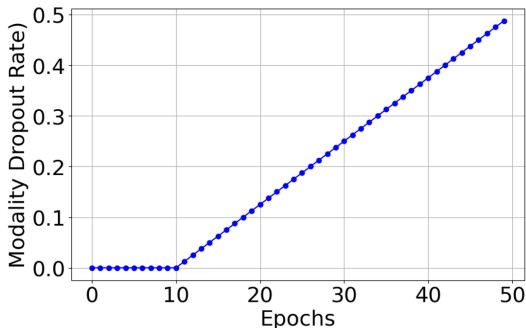

Figure 11: Modality Dropout Scheme based on Curriculum Learning (supporting illustration for Section 3.2.5).

Table 6: Model design components and hyperparameters.

| Design Component | Hyperparameter | Value |
|---|---|---|
| Symbolic Tokenization | $\alpha$ – Number of alphabets | 20 |
| | $W$ – Compression factor | 2 |
| Max Attention Budget | $\beta$ | 5 |
| Sparse MoE | $\Omega$ – Number of experts | 4 |
| | $k$ – Selected experts per token | 1 |

**Macro-F1.** Let $\mathcal{C}$ be the set of classes. For each class $c \in \mathcal{C}$, we compute precision $P_c$ and recall $R_c$:

$$\text{F1}_c = \frac{2P_c R_c}{P_c + R_c}, \quad \text{Macro-F1} = \frac{1}{|\mathcal{C}|} \sum_{c \in \mathcal{C}} \text{F1}_c$$

**Relative Improvement.** We report relative measurements for all metrics as follows. Let $\text{Metric}_{\text{base}}$ be the baseline performance and $\text{Metric}_{\text{ours}}$ be our model's performance. The relative improvement is:

$$\text{Relative Improvement} = \frac{\text{Metric}_{\text{ours}} - \text{Metric}_{\text{base}}}{\text{Metric}_{\text{base}}} \times 100\%$$

**Absolute Improvement.** We report absolute improvement in the case of accuracy-based metrics as the direct difference between our model's performance and the baseline performance, i.e.:

$$\text{Absolute Improvement} = \text{Metric}_{\text{ours}} - \text{Metric}_{\text{base}}$$

### D.2 Dataset Details

In this section, we present detailed information about the datasets used for evaluation, including class distributions, overall statistics, and preprocessing steps.

**WESAD.** We directly leverage the synchronized data from [46] for the WESAD dataset. The modalities and their corresponding sampling rates are summarized in Table 7, and the class distribution is shown in Figure 12.

**DaliaHAR.** We adapt the DaLiA dataset for activity recognition using multimodal sensor data, addressing the scarcity of datasets that offer both diverse modalities and fine-grained activity labels. Figure 13 illustrates the raw distribution of activity labels from a single subject recording. To preprocess the data, we first segment the continuous recordings based on absolute timestamps provided in the annotation files. Non-informative segments such as baseline and no-activity periods are excluded. We then apply a sliding window approach with a window size of 8 seconds and a 2-second overlap. Since all sensor streams are temporally aligned, each modality is segmented consistently, and each window is assigned the corresponding activity label for that subject.

The resulting processed data summary is given in Table 8 and the class distribution is shown in Figure 14.

Table 7: WESAD dataset modality details.

| Modality | Sampling Rate (Hz) | Variates |
|---|---|---|
| chest_ACC | 700 | 3 |
| chest_ECG | 700 | 1 |
| chest_EMG | 700 | 1 |
| chest_RESP | 700 | 1 |
| chest_EDA | 700 | 1 |
| chest_TEMP | 700 | 1 |
| wrist_ACC | 32 | 3 |
| wrist_BVP | 64 | 1 |
| wrist_EDA | 4 | 1 |
| wrist_TEMP | 4 | 1 |
| *Output Classes: Baseline, Stress, Amusement* | | |

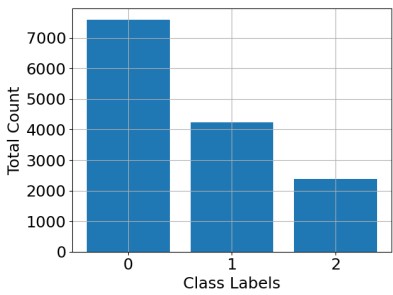

Figure 12: Distribution of the classes for WESAD dataset.

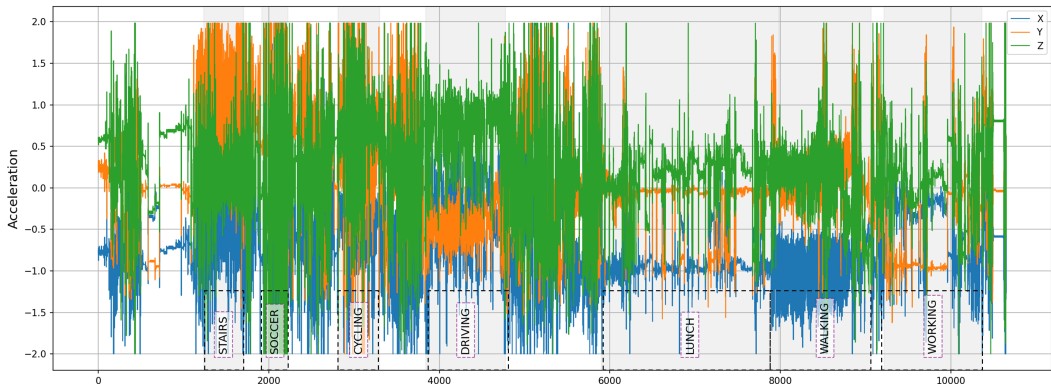

Figure 13: Visualizing the raw accelerometer data from wrist for the Dalia dataset.

| Modality | Sampling Rate (Hz) | Variates |
|---|---|---|
| chest_ACC | 700 | 3 |
| wrist_ACC | 32 | 3 |
| wrist_BVP | 64 | 1 |
| wrist_EDA | 4 | 1 |
| wrist_TEMP | 4 | 1 |
| *Output Classes: STAIRS, SOCCER, CYCLING, DRIVING, LUNCH, WALKING, WORKING* | | |

Table 8: DaliaHAR dataset modality details.

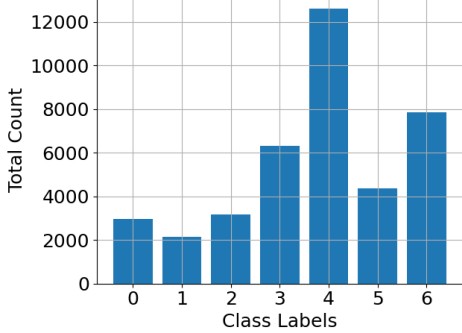

Figure 14: Distribution of activity classes in the DaliaHAR dataset.

**DSADS.** For the DSADS dataset, we follow the original preprocessing steps and report the corresponding sensor modalities along with their specifications in Table 9. The class distribution is presented in Figure 15.

**MIMIC.** We adopt the preprocessing for the MIMIC dataset as defined by the multimodal benchmarking suite, MultiBench [25], to comply with standardized benchmarking practices. Overall, our performance—shown in Table 2 in Section 4.1 of the main paper—aligns with the results reported in the benchmarking suite. However, we observe a clear class imbalance, as illustrated in Figure 16. Therefore, we also report the Macro-F1 score, which is particularly more informative than accuracy for the MIMIC dataset.

| Modality | Sampling Rate (Hz) | Variates |
|---|---|---|
| torso | 25 | 9 |
| right_arm | 25 | 9 |
| left_arm | 25 | 9 |
| right_leg | 25 | 9 |
| left_leg | 25 | 9 |
| *Output Classes: Sitting, Standing, Lying on back, Lying on right side, Ascending stairs, Descending stairs, Standing in an elevator (still), Moving around in an elevator, Walking in a parking lot, Walking on treadmill (flat), Walking on treadmill (inclined), Running on treadmill, Exercising on a stepper, Exercising on a cross trainer, Cycling on exercise bike (horizontal), Cycling on exercise bike (vertical), Rowing, Jumping, Playing basketball* | | |

Table 9: DSADS dataset modality details.

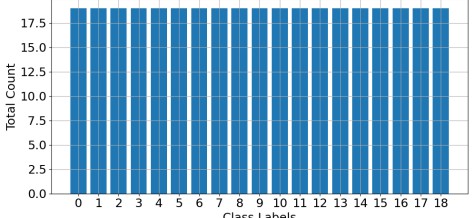

Figure 15: Class distribution of DSADS activity labels.

| Modality | Sampling Rate (Hz) | Variates |
|---|---|---|
| glasgow | 1 | 1 |
| BP | 1 | 1 |
| HR | 1 | 1 |
| Temp | 1 | 1 |
| oxy | 1 | 1 |
| urine | 1 | 1 |
| urea | 1 | 1 |
| wbc | 1 | 1 |
| bdc2 | 1 | 1 |
| Na | 1 | 1 |
| K | 1 | 1 |
| Bil | 1 | 1 |
| Age | 1 | 1 |
| icd9 | 1 | 1 |
| hem_mal | 1 | 1 |
| cancer | 1 | 1 |
| adm_type | 1 | 1 |
| *Output Classes: 6 ICD-9 diagnostic categories (coarse-grained)* | | |

Table 10: MIMIC-III dataset modality details.

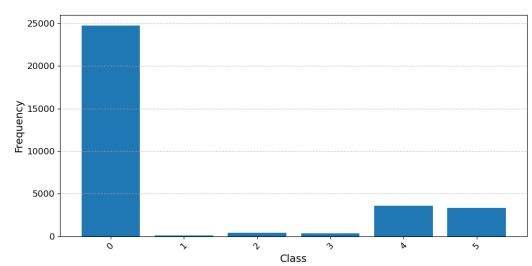

Figure 16: Distribution of ICD-9 class labels in the MIMIC-III dataset.

## D.3 Baseline Implementation Details

**Multivariate Baselines.** We follow the MTS-Bakeoff [37] implementation for InceptionTime and ResNet1D. For the Transformer, we adopt the standard implementation with 8 heads and 2 layers, including positional encoding. In the case of the missingness analysis in Figure 5 of the main paper, we adapt the Transformer by retraining it using the same modality dropout scheme as employed in MAESTRO, to enable a fair comparison with a multivariate baseline that is not natively robust to missing data. This modification allows the adapted Transformer to serve as a competitive baseline, as evidenced by the performance trends shown in Figure 5.

**Multimodal Baselines.** For the LRTF and MULT baselines, we implement them following the MultiBench [25] framework. For the remaining baselines, we use their original implementations and open-source resources for reproduction.

In the robustness study in Section 4.1 in the main paper, we include only those baselines that natively support missingness, along with the adapted Transformer. We train these models using their original settings and evaluate them on samples with dynamically missing modalities. Specifically, we randomly drop out modalities with increasing severity, ranging from 10% to 40% and report the performance.

# E More Detailed Results of the Main Paper Experiments

This section includes more detailed results from those experiments discussed in Section 4 of the main paper.

## E.1 Robustness Results

The primary results using the Macro-F1 performance metric for varying levels of missingness across all datasets are presented in Figure 5 in Section 4.1 of the main paper. Tables 11, 12, 9, and 14 provide the complete statistics for accuracy and F1 scores for all datasets—WESAD, DaliaHAR, DSADS, and MIMIC-III, respectively.

Table 11: Accuracy and F1-score (mean$_{std}$) across different missingness levels for the WESAD dataset (supporting results for Figure 5 in Section 4.1 in the main paper).

| Model | 0% | | 10% | | 20% | | 30% | | 40% | |
|---|---|---|---|---|---|---|---|---|---|---|
| | Acc | F1 | Acc | F1 | Acc | F1 | Acc | F1 | Acc | F1 |
| Transformer | $0.67_{0.04}$ | $0.53_{0.06}$ | $0.63_{0.09}$ | $0.45_{0.06}$ | $0.62_{0.00}$ | $0.49_{0.05}$ | $0.61_{0.04}$ | $0.48_{0.09}$ | $0.61_{0.08}$ | $0.46_{0.01}$ |
| FlexMoE | $0.71_{0.09}$ | $0.64_{0.11}$ | $0.68_{0.08}$ | $0.61_{0.08}$ | $0.65_{0.06}$ | $0.58_{0.09}$ | $0.62_{0.08}$ | $0.56_{0.08}$ | $0.59_{0.07}$ | $0.52_{0.07}$ |
| FuseMoE | $0.48_{0.13}$ | $0.40_{0.06}$ | $0.46_{0.01}$ | $0.41_{0.03}$ | $0.45_{0.01}$ | $0.40_{0.03}$ | $0.41_{0.03}$ | $0.38_{0.04}$ | $0.39_{0.03}$ | $0.37_{0.02}$ |
| ShaSpec | $0.65_{0.56}$ | $0.54_{0.44}$ | $0.55_{0.51}$ | $0.49_{0.44}$ | $0.53_{0.50}$ | $0.47_{0.44}$ | $0.48_{0.46}$ | $0.44_{0.41}$ | $0.43_{0.42}$ | $0.41_{0.39}$ |
| Ours(w/o SAX) | $0.69_{0.13}$ | $0.55_{0.06}$ | $0.68_{0.11}$ | $0.54_{0.05}$ | $0.67_{0.09}$ | $0.53_{0.04}$ | $0.65_{0.08}$ | $0.52_{0.04}$ | $0.66_{0.08}$ | $0.53_{0.06}$ |
| Ours | $0.77_{0.07}$ | $0.66_{0.03}$ | $0.77_{0.06}$ | $0.68_{0.04}$ | $0.74_{0.07}$ | $0.54_{0.04}$ | $0.74_{0.07}$ | $0.64_{0.06}$ | $0.71_{0.07}$ | $0.61_{0.06}$ |

Table 12: Accuracy and F1-score (mean$_{std}$) across different missingness levels for the DaliaHAR dataset (supporting results for Figure 5 in Section 4.1 in the main paper).

| Model | 0% | | 10% | | 20% | | 30% | | 40% | |
|---|---|---|---|---|---|---|---|---|---|---|
| | Acc | F1 | Acc | F1 | Acc | F1 | Acc | F1 | Acc | F1 |
| Transformer | $0.72_{0.04}$ | $0.70_{0.08}$ | $0.69_{0.04}$ | $0.67_{0.07}$ | $0.67_{0.04}$ | $0.65_{0.07}$ | $0.64_{0.06}$ | $0.62_{0.08}$ | $0.61_{0.06}$ | $0.59_{0.08}$ |
| FlexMoE | $0.70_{0.06}$ | $0.70_{0.05}$ | $0.63_{0.04}$ | $0.61_{0.05}$ | $0.53_{0.04}$ | $0.51_{0.06}$ | $0.45_{0.01}$ | $0.43_{0.02}$ | $0.37_{0.03}$ | $0.35_{0.03}$ |
| FuseMoE | $0.78_{0.01}$ | $0.79_{0.03}$ | $0.67_{0.01}$ | $0.68_{0.03}$ | $0.58_{0.01}$ | $0.58_{0.03}$ | $0.48_{0.03}$ | $0.46_{0.03}$ | $0.40_{0.04}$ | $0.36_{0.03}$ |
| ShaSpec | $0.74_{0.00}$ | $0.77_{0.02}$ | $0.64_{0.02}$ | $0.67_{0.03}$ | $0.55_{0.02}$ | $0.57_{0.02}$ | $0.48_{0.02}$ | $0.47_{0.01}$ | $0.44_{0.02}$ | $0.39_{0.01}$ |
| Ours(w/o SAX) | $0.82_{0.03}$ | $0.84_{0.03}$ | $0.80_{0.05}$ | $0.83_{0.02}$ | $0.77_{0.04}$ | $0.79_{0.03}$ | $0.73_{0.03}$ | $0.76_{0.02}$ | $0.71_{0.02}$ | $0.73_{0.01}$ |
| Ours | $0.83_{0.01}$ | $0.84_{0.01}$ | $0.83_{0.04}$ | $0.85_{0.03}$ | $0.81_{0.03}$ | $0.82_{0.03}$ | $0.78_{0.03}$ | $0.79_{0.03}$ | $0.74_{0.03}$ | $0.75_{0.03}$ |

Table 13: Accuracy and F1-score (mean$_{std}$) across different missingness levels for the DSADS dataset (supporting results for Figure 5 in Section 4.1 in the main paper).

| Model | 0% | | 10% | | 20% | | 30% | | 50% | |
|---|---|---|---|---|---|---|---|---|---|---|
| | Acc | F1 | Acc | F1 | Acc | F1 | Acc | F1 | Acc | F1 |
| Transformer | $0.88_{0.04}$ | $0.88_{0.04}$ | $0.78_{0.13}$ | $0.77_{0.15}$ | $0.74_{0.16}$ | $0.73_{0.14}$ | $0.71_{0.13}$ | $0.70_{0.15}$ | $0.66_{0.10}$ | $0.68_{0.09}$ |
| FlexMoE | $0.67_{0.01}$ | $0.63_{0.03}$ | $0.46_{0.05}$ | $0.44_{0.05}$ | $0.35_{0.09}$ | $0.34_{0.08}$ | $0.25_{0.00}$ | $0.23_{0.01}$ | $0.19_{0.06}$ | $0.17_{0.07}$ |
| FuseMoE | $0.84_{0.03}$ | $0.85_{0.03}$ | $0.59_{0.00}$ | $0.61_{0.01}$ | $0.46_{0.04}$ | $0.46_{0.04}$ | $0.43_{0.02}$ | $0.44_{0.03}$ | $0.37_{0.03}$ | $0.38_{0.03}$ |
| ShaSpec | $0.81_{0.01}$ | $0.79_{0.00}$ | $0.58_{0.03}$ | $0.58_{0.03}$ | $0.45_{0.02}$ | $0.45_{0.02}$ | $0.29_{0.01}$ | $0.27_{0.02}$ | $0.22_{0.01}$ | $0.20_{0.02}$ |
| Ours(w/o SAX) | $0.78_{0.02}$ | $0.77_{0.02}$ | $0.79_{0.00}$ | $0.77_{0.01}$ | $0.76_{0.01}$ | $0.75_{0.01}$ | $0.72_{0.01}$ | $0.71_{0.01}$ | $0.66_{0.05}$ | $0.65_{0.04}$ |
| Ours | $0.89_{0.01}$ | $0.88_{0.01}$ | $0.86_{0.01}$ | $0.86_{0.01}$ | $0.84_{0.00}$ | $0.84_{0.01}$ | $0.83_{0.01}$ | $0.82_{0.01}$ | $0.79_{0.02}$ | $0.79_{0.02}$ |

Table 14: Accuracy and F1-score (mean$_{std}$) across different missingness levels in MIMIC dataset (Supporting results for Figure 5 in Section 4.1 in the main paper).

| Model | 0% | | 10% | | 20% | | 30% | | 40% | |
|---|---|---|---|---|---|---|---|---|---|---|
| | Acc | F1 | Acc | F1 | Acc | F1 | Acc | F1 | Acc | F1 |
| Transformer | $0.78_{0.02}$ | $0.22_{0.01}$ | $0.78_{0.04}$ | $0.21_{0.03}$ | $0.78_{0.01}$ | $0.22_{0.02}$ | $0.77_{0.01}$ | $0.22_{0.01}$ | $0.77_{0.04}$ | $0.21_{0.03}$ |
| FlexMoE | $0.79_{0.02}$ | $0.27_{0.03}$ | $0.78_{0.02}$ | $0.25_{0.02}$ | $0.79_{0.01}$ | $0.25_{0.01}$ | $0.78_{0.01}$ | $0.24_{0.02}$ | $0.78_{0.01}$ | $0.21_{0.01}$ |
| FuseMoE | $0.76_{0.01}$ | $0.26_{0.00}$ | $0.75_{0.01}$ | $0.27_{0.01}$ | $0.73_{0.02}$ | $0.23_{0.00}$ | $0.72_{0.03}$ | $0.23_{0.01}$ | $0.72_{0.02}$ | $0.22_{0.01}$ |
| ShaSpec | $0.76_{0.01}$ | $0.25_{0.01}$ | $0.74_{0.03}$ | $0.25_{0.01}$ | $0.74_{0.01}$ | $0.23_{0.00}$ | $0.74_{0.00}$ | $0.20_{0.02}$ | $0.70_{0.05}$ | $0.21_{0.00}$ |
| Ours(w/o SAX) | $0.80_{0.01}$ | $0.31_{0.01}$ | $0.80_{0.02}$ | $0.31_{0.02}$ | $0.77_{0.01}$ | $0.28_{0.01}$ | $0.79_{0.02}$ | $0.27_{0.01}$ | $0.77_{0.02}$ | $0.26_{0.02}$ |
| Ours | $0.79_{0.01}$ | $0.30_{0.02}$ | $0.78_{0.02}$ | $0.30_{0.03}$ | $0.76_{0.01}$ | $0.29_{0.03}$ | $0.78_{0.01}$ | $0.27_{0.01}$ | $0.77_{0.02}$ | $0.25_{0.02}$ |

## E.2 Ablation Results

The complete statistics of the Figure 6 in Section 4.2 of the main paper is given in Table 15.

Additional results on the sensitivity to hyperparameters $\frac{T}{W}$ and $\Omega$ experts in the Sparse-MoE layer, across two compute platforms RTX A6000 and Jetson TX2.

## E.3 Simple Additive Noise

In the current work, we are focused on handling complete modality missings, which is a common issue in real-world sensing applications with a large number of modalities ($N > 4$), where a sensor

Table 15: Accuracy (mean$_{std}$) on the WESAD dataset under full data and 40% missingness conditions for an ablation analysis (supporting results for Figure 6 in Section 4.2 of the main paper).

| Method | Full | 40% Missing |
|---|---|---|
| Ours | $0.78_{0.07}$ | $0.71_{0.06}$ |
|    w/o Symbolic Transformation | $0.69_{0.13}$ | $0.66_{0.07}$ |
|    w/o Modality Embedding | $0.70_{0.10}$ | $0.55_{0.04}$ |
|    w/o Modality Dropout | $0.76_{0.09}$ | $0.62_{0.07}$ |
|    w/o Adaptive Attn Budget | $0.71_{0.11}$ | $0.67_{0.08}$ |
|    w/o Cross-modal SparseMoE | $0.70_{0.08}$ | $0.50_{0.01}$ |

Table 16: Performance comparison across number of experts.

| # Experts | ACC | RTX A6000 | | Jetson TX2 | |
|---|---|---|---|---|---|
| | | Latency (ms) | GFLOPs | Latency (ms) | GFLOPs |
| 8 | 0.80 | 352.09 | 7.15 | 1298.52 | 6.16 |
| 4 | 0.79 | 279.41 | 7.08 | 1315.08 | 6.12 |
| 2 | 0.75 | 245.03 | 7.07 | 1291.25 | 6.00 |

fails and we are left with an arbitrary set of multimodal inputs to make predictions. We also include a preliminary study on noise robustness, which focuses on robustness to missing modalities, it could be a promising direction for future work on MAESTRO. We have conducted some initial exploration with three types of noise defined as follows. Here $\tilde{x}$ denotes the subset of modalities chosen for introducing corruption:

1. **Random noise**: Selected channels are replaced with i.i.d. Gaussian noise, $\tilde{x} = \epsilon$, $\epsilon \sim \mathcal{N}(0, \sigma^2)$,    for $c \in \mathcal{C}$

2. **Additive noise**: Gaussian noise is added to the original signal, $\tilde{x} = x + \epsilon$, $\epsilon \sim \mathcal{N}(0, \sigma^2)$, for $c \in \mathcal{C}$

3. **Additive noise with spikes** (representing electrical spikes): Sparse high-magnitude impulses (spikes) are added in addition to Gaussian noise:

$$\tilde{x} = x + \epsilon + \mathbf{M} \odot \mathbf{S}, \quad \epsilon \sim \mathcal{N}(0, \sigma^2)$$

where $\mathbf{M} \sim \text{Bernoulli}(p)$ is a binary mask indicating spike locations with $p$ probability, $\mathbf{S} \in \{-m, +m\}$ is the spike magnitude, $\odot$ denotes element-wise multiplication.

We present the zero-shot performance of MAESTRO on noisy WESAD data in the following two combinations: in combination 1, we dropped the chest accelerometer, and in combination 2, we dropped the wrist BVP, EDA, and Temp. Please refer to Table 6 in our Appendix for more details on the WESAD modalities.

Also, in Figure 8, the Symbolic representation without compression refers to only using a word length of 1 and only quantizing the time-series values to a fixed vocabulary.

### E.4   Pilot experiments on Asynchronous Modalities

We present a brief pilot study in a simplified setting to emulate asymnchronous input data or partial-missingness of a modality. We mask approximately 25% of the samples from the beginning and end of the sequence in 20% of the modalities (in this case, the wrist BVP, TEMP, and EDA) during inference to demonstrate zero-shot transfer capability. Currently, the model treats this partial missingness similarly or slightly better than complete missingness.

This performance could potentially be improved by incorporating standard techniques, such as absolute positional encoding, or by leveraging prior models like mTAN [49], which can more effectively encode irregular time-series with semantic meaning to support downstream cross-modal learning. Additionally, the generalization of the MAESTRO framework to distribution shifts—possibly arising from upgrades to one of the sensing modalities—could be enhanced using low-overhead design strategies such as PhASER [41].

Table 17: Performance comparison across varying word lengths for SAX.

| Word Length | ACC | RTX A6000 | | Jetson TX2 | |
|---|---|---|---|---|---|
| | | Latency (ms) | GFLOPs | Latency (ms) | GFLOPs |
| 2 | 0.80 | 279.41 | 7.08 | 1315.10 | 6.12 |
| 4 | 0.77 | 270.00 | 3.56 | 677.81 | 3.06 |
| 8 | 0.80 | 275.69 | 1.74 | 349.33 | 1.50 |

Table 18: Zero-shot transfer accuracy for MAESTRO under partial modality missingness scenarios.

| Modalities | Accuracy |
|---|---|
| All available | 0.77 |
| Completely missing 20% modalities | 0.61 |
| Mask first 25% time steps in 20% modalities | 0.60 |
| Mask last 25% time steps in 20% modalities | 0.65 |

### E.5 High density input samples

We resample the modalities in WESAD to increase the sequence length in order to emulate high-frequency sensor readings and report the performance below. As expected, our proposed MAESTRO performs consistently even with longer sequence lengths. However, upon replacing the sparse attention with canonical dense self-attention layers in both the intra-modal and cross-modal stages, we encounter out-of-memory (OOM) issues at higher sampling rates (in this case, 128 Hz), highlighting MAESTRO's advantage in resource efficiency. All experiments were conducted on the RTX A6000.

Table 19: Comparison of MAESTRO and dense attention models under different sampling rates. OOM = out-of-memory.

| Sampling Rate | MAESTRO | | | Dense Attention | | |
|---|---|---|---|---|---|---|
| | Acc | GFLOPs | MMACs | Acc | GFLOPs | MMACs |
| 32 | 0.77 | 6.13 | 3066 | 0.75 | 8.78 | 4205 |
| 64 | 0.77 | 14.41 | 7205 | 0.71 | 23.02 | 11510 |
| 128 | 0.73 | 29.01 | 14502 | OOM | OOM | OOM |

## F Additional Experiments

In additional to the experiments shown in the main paper and Section E, we also conducted additional experiments to further evaluate our approach, as shown below.

### F.1 Unimodal Sweep Results

This section presents the unimodal results for WESAD, DSADS, DALIAHAR, and MIMIC, as shown in Table 20. For unimodal training, we train individual models for each modality using the Transformer backbone. These results indicate that in some applications, the modalities contain redundant information—as seen in DSADS—where unimodal performance is not significantly lower than multimodal performance. In contrast, in datasets like Dalia, certain modalities perform close to random guessing (e.g., `wrist_TEMP` with 0.38 accuracy for a 3-class classification task). Since MIMIC shows overall lower performance, we report the top five modalities and additionally provide the F1-score to highlight the performance boost achieved by using `MAESTRO` (with an F1-score of 0.30), compared to unimodal models which typically achieve around 0.15 F1 in most cases.

### F.2 Supporting Results for Case Study on DaliaHAR from Section 4.2

Based on a known *a priori*, we evaluate three scenarios: 1) using the two best modalities—`wrist_ACC` and `chest_ACC`—we train a bimodal model, 2) we run inference on the adapted Transformer using only these two modalities while dropping the rest, and 3) we evaluate `MAESTRO` in the presence of only these two modalities.

Table 20: Unimodal performance across all datasets (supporting results for Section 4.2).

| Dataset | Modality | ACC | STDEV |
|---------|----------|-----|-------|
| DSADS | Torso | 0.63 | 0.01 |
| | Right Arm | 0.74 | 0.02 |
| | Left Arm | 0.85 | 0.03 |
| | Right Leg | 0.81 | 0.02 |
| | Left Leg | 0.83 | 0.01 |
| Dalia | wrist_ACC | 0.81 | 0.04 |
| | wrist_BVP | 0.50 | 0.04 |
| | wrist_EDA | 0.45 | 0.03 |
| | wrist_TEMP | 0.35 | 0.03 |
| | chest_ACC | 0.85 | 0.01 |
| WESAD | chest_ACC | 0.67 | 0.04 |
| | chest_ECG | 0.75 | 0.13 |
| | chest_EMG | 0.75 | 0.01 |
| | chest_RESP | 0.63 | 0.04 |
| | chest_EDA | 0.60 | 0.04 |
| | chest_TEMP | 0.71 | 0.02 |
| | wrist_ACC | 0.66 | 0.12 |
| | wrist_BVP | 0.71 | 0.08 |
| | wrist_EDA | 0.60 | 0.04 |
| | wrist_TEMP | 0.38 | 0.12 |
| MIMIC | glasgow | 0.77 | 0.01 |
| | BP | 0.72 | 0.08 |
| | HR | 0.72 | 0.07 |
| | Temp | 0.77 | 0.01 |
| | oxy | 0.76 | 0.01 |

Table 21: F1 score and standard deviation for 5 best MIMIC modalities.

| Modality | F1 Score | STD |
|----------|----------|-----|
| glasgow | 0.17 | 0.02 |
| BP | 0.17 | 0.02 |
| HR | 0.16 | 0.02 |
| Temp | 0.15 | 0.01 |
| oxy | 0.16 | 0.01 |

Our results, shown in Table 22, highlight that MAESTRO with *a priori* knowledge at run-time performs competitively (with an absolute improvement of 3%) compared to an end-to-end model trained on these pre-selected modalities. This demonstrates that MAESTRO is capable of effectively modeling multimodal interactions and learning task-dependent semantics from multimodal data.

Table 22: Accuracy with *a priori* modality mask at inference: [wrist_ACC, wrist_BVP, wrist_EDA, wrist_TEMP, chest_ACC] = [1, 0, 0, 0, 1] (supporting results for Section 4.2).

| Method | Accuracy |
|--------|----------|
| Bimodal (wrist + chest) | $0.82_{\pm 0.02}$ |
| Transformer (adapted) | $0.71_{\pm 0.01}$ |
| MAESTRO | $0.85_{\pm 0.01}$ |

## F.3 Visualization of Unimodal Representations for DaliaHAR

To further support the results in Table 5, we plot the t-SNE projections of the unimodal models' latent representations for each modality in DaliaHAR, as shown in Figure 17.

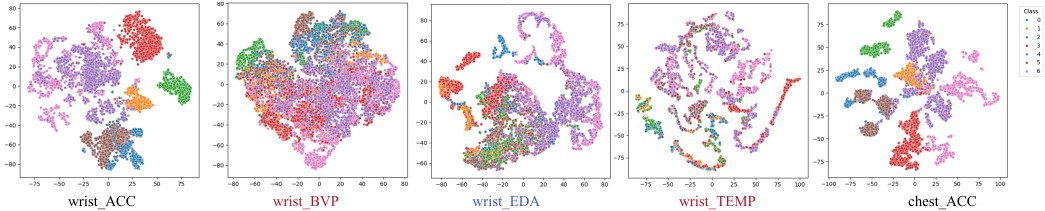

Figure 17: t-SNE projections of the unimodal models' latent representations for each modality in DaliaHAR (supporting results for Table 5 in Section 4.2 in the main paper).

# G Broader Impacts

`MAESTRO` paves the way for more efficient and practical handling of heterogeneous sensing data. It has the potential to enhance analytics and, in turn, the performance of ubiquitous sensing applications across diverse domains—including smart home monitoring, daily living assistance, fitness and wellness interventions, elderly care, healthcare, and environmental monitoring. These applications rely on rich, continuous streams of sensory data, where sensor reliability often cannot be guaranteed. For example, in environmental monitoring within remote or inaccessible locations, sensors may fail due to power loss or harsh conditions. In such scenarios, maintaining robust performance with only a subset of available modalities is critical.

Currently, `MAESTRO` addresses complete modality-level missingness, demonstrating its ability to sustain model effectiveness even under challenging sensing conditions. In future, we aim to explore more advanced symbolic encoding strategies and extend the framework to address irregularly sampled and asynchronous sensing modalities and

