# OpenReview forum: "MAESTRO : Adaptive Sparse Attention and Robust Learning for Multimodal Dynamic Time Series"
_NeurIPS.cc/2025/Conference — NeurIPS 2025 spotlight_

### Official Review · Reviewer_bCcz · 2025-06-27

**Clarity:** 3
**Significance:** 3
**Originality:** 3
**Rating:** 4
**Confidence:** 3

**Summary:**

MAESTRO is a multimodal learning framework designed for dynamic time-series data from heterogeneous sensors, addressing challenges like missing modalities and inefficient pairwise modeling. It uses symbolic tokenization to represent time-series data compactly while explicitly marking missing inputs. An adaptive sparse attention mechanism allocates attention based on modality relevance, and a sparse mixture-of-experts layer enables dynamic routing depending on available modalities. Experiments on four real-world datasets show that MAESTRO outperforms state-of-the-art baselines in both accuracy and robustness, especially under partial modality settings.

**Questions:**

1. The framework is only evaluated on classification problems. What about its applicability to other tasks of different type?
2. Although the proposed method handles missing modalities well, the model's behavior under noisy inputs is not systematically evaluated. I am curious about its robustness to the noisy data.
3. One minor issue: "reliance on" in line 5 should be put ahead of "(1)"?

**Ethical Concerns:**

["NO or VERY MINOR ethics concerns only"]

**Final Justification:**

The authors have addressed my concerns. I have no more concern.

**Limitations:**

Yes.

**Paper Formatting Concerns:**

None.

**Quality:**

3

**Strengths And Weaknesses:**

Strengths

- MAESTRO is robust to missing modalities. It handles arbitrary combinations of available sensor modalities using symbolic tokenization and dynamic routing, outperforming baselines.
- By employing sparse attention and avoiding pairwise interaction modeling, MAESTRO significantly reduces computational cost compared to dense multimodal methods like MULT.

Weaknesses

- It seems that the model involves multiple components that may require extensive hyperparameter tuning for the new datasets.

---

> ### Author Rebuttal · Authors · 2025-07-31
>
> Thank you for the encouraging review and the recognition of MAESTRO's performance and efficiency under practical settings of learning from heterogenous time-series modalities. We provide our response below :
>
> > It seems that the model involves multiple components that may require extensive hyperparameter tuning for the new datasets.
>
> In our current work we have not extensively fine-tuned our hyper-parameters ( (as shown in Table 5 of the Appendix), yet we are able to outperform the existing baselines under complete and arbitrary modality combinations. However, for application-specific scenarios, based on the available computational resources, these hyperparameters can be selected to optimize for performance and computational efficiency. We present a brief sensitivity analyses for number of experts and the word length for the WESAD dataset across two computational platforms below :
>
> ##### Varying Experts for MoE (default w = 2)
>
>
> | # Experts | ACC | RTX A6000 - Latency (ms) | RTX A6000 - GFLOPs  | Jetson TX2 - Latency (ms)   | Jetson TX2 - GFLOPs |
> |-----------|----------|--------------------------|---------------------|-----------------------------|---------------------|
> | 8		    |   0.80   |     352.09	              |       7.15          |	 1298.52	              |      6.16           |
> | 4		    |   0.79   |     279.41	              |       7.08          |	 1315.08	              |      6.12           |
> | 2		    |   0.75   |     245.03	              |       7.07          |	 1291.25	              |      6              |
>
>
> ##### Varying Word Length for SAX (default w = 2)
>
> | Word Length (w) | ACC | RTX A6000 - Latency (ms) | RTX A6000 - GFLOPs | Jetson TX2 - Latency (ms) | Jetson TX2 - GFLOPs |
> |------------------|----------|---------------------------|----------------------|-----------------------------|-----------------------|
> |2		           |  0.80    |  279.41                   |	  7.08	             |        1315.08	            |       6.12           |
> |4		           |  0.77    |  270.00                   |	  3.56	             |        677.81	             |      3.06           |
> |8		           |  0.80    |  275.69                   |	  1.74	             |        349.33	             |      1.5           |
>
>
> Thank you for commenting on this aspect. We will include this analyses in our Appendix.
>
>
>
> > MAESTRO is evaluated on classification tasks; discuss applicability to other tasks.
>
> We currently explore classification tasks as they are the commonly occuring multimodal tasks (activity recognition, clinical disease identification, stress detection etc.) in consumer and clinical healthcare and lifestyle applications involving multiple heterogenous sensors collecting time-series data, which is the focus of our work. Also current multimodal methods with time-series are designed for predictive applications allowing us to benchmark MAESTRO.
>
> We think, fundamentally MAESTRO's arechitecture can be extended to support generative tasks with mulitmodal input. In future work, we plan to explore two key directions for such extension: 1) curating a multimodal generative tasks dataset; and 2) exploring encoder-only vs. encoder-decoder cross-attention architectures for generation. One of the generative application spaces that stands to benefit from cross-modal attention to heterogeneous modalities without resource efficiency is robot manipulation. Recent works [1, 2] in robot manipulation explore enriching the latent representations of the action sequence through multimodal inputs before learning the Reinforcement-Learning policy. Often, there are time-series modalities [3, 4] that are availabile arbitrarily in these applications. We plan to explore the applicability of MAESTRO to such settings (to learn better action-sequence representation from multimodal inputs) in future.
>
> We will add this discussion to our "Broader Impact and Future Work".
>
>
> [1] Decision Transformer: Reinforcement Learning via Sequence Modeling, 2021
>
> [2] A Vision-Language-Action Flow Model for General Robot Control, 2025
>
> [3] MimicTouch: Leveraging Multi-modal Human Tactile Demonstrations for Contact-rich Manipulation, 2024
>
> [4] The MOTIF Hand: A Robotic Hand for Multimodal Observations with Thermal, Inertial, and Force Sensors, 2025
>
> ---
>
> > While MAESTRO handles missing data well, the model's behavior under noisy inputs is not systematically evaluated. I am curious about its robustness to the noisy data.
>
> In the current work, we are focused on handling complete modality missingness, which is a common issue in real-world sensing applications with a large number of modalities (N > 4), where a sensor fails and we are left with an arbitrary set of multimodal inputs to make predictions, as mentioned in our paper's scope (lines 10, 62, 74).
>
> While an extensive study on noise robustness is out of scope for this paper, which focuses on robustness to missing modalities, it could be a promising direction for future work on MAESTRO. We have just conducted some initial exploration with three types of noise defined as follows. Here $\tilde{\mathbf{x}}$ denotes the subset of modalities chosen for introducing corruption or omission :
>
> 1. Random noise : Selected channels are replaced with i.i.d. Gaussian noise, $\tilde{\mathbf{x}} = \boldsymbol{\epsilon}, \quad \boldsymbol{\epsilon} \sim \mathcal{N}(0, \sigma^2), \quad \text{for } c \in \mathcal{C}$
> 2. Additive noise  :  Gaussian noise is added to the original signal, $\tilde{\mathbf{x}} = \mathbf{x} + \boldsymbol{\epsilon}, \quad \boldsymbol{\epsilon} \sim \mathcal{N}(0, \sigma^2), \quad \text{for } c \in \mathcal{C}$.
> 3. Additive noise with spikes (representing electrical spikes) : Sparse high-magnitude impulses (spikes) are added in addition to Gaussian noise: $\tilde{\mathbf{x}} = \mathbf{x} + \boldsymbol{\epsilon} + \mathbf{M} \odot \mathbf{S}, \quad \boldsymbol{\epsilon} \sim \mathcal{N}(0, \sigma^2)$,
>    where $\mathbf{M} \sim \text{Bernoulli}(p)$ is a binary mask indicating spike locations  with $p$ probability, $\mathbf{S} \in \{-m, +m\}$ is the spike magnitude, $\odot$ denotes element-wise multiplication
>
>
>
> We present the zero-shot performance of MAESTRO on noisy WESAD data in the following two combinations : in combination 1, we dropped the chest accelerometer, and in combination 2, we dropped the wrist BVP, EDA, and Temp. Please refer to Table 6 in our Appendix for more details on the WESAD modalities.
>
> | Corruption Type            | Combination 1              | Combination 2                       |
> |----------------------------|---------------------------|-----------------------------------|
> | *None (Clean and Complete Modalities)*               | *0.77*                      | *0.77*                              |
> | Missing                    | 0.61                      | 0.65                              |
> | Random Noise               | 0.61                      | 0.63                              |
> | Additive Noise (σ = 0.5)   | 0.75                      | 0.74                              |
> | Additive Noise + Spikes (σ = 0.5, p = 0.1, m = 5)   | 0.60                      | 0.73                              |
>
> We observe that for simple noise settings like random noise, MAESTRO treats those modalities similar to complete missingness of that modality.
>
> In the simple additive noise setting, we are able to perform almost as well as the case where all modalities are present. This is not surprising because (1) our SAX representation provides robustness to local perturbation, and (2) our cross-modal attention allows querying only for the relevant inter- and intra-modal values relevant to the task.
>
> Finally, a more aggressive noise where we simulate electrical noise interference as spikes shows the same performance in one combination as missingness and in another combination very close to the full modality scenario.
>
> We can consider missingness to be one of the aggressive noise corruptions, and through this controlled experiment, we can see MAESTRO's robustness to simple perturbations. Thank you for raising this aspect of MAESTRO's robustness to noisy inputs. We will include the above preliminary analyses in our Appendix to motivate and guide potential future exploration, e.g., to explore other systematic noise injection mechanisms and their impact on MAESTRO.
>
> ---
>
> > Minor writing issue: "reliance on" in line 5 should be put ahead of "(1)"?
>
> We will update our abstract to reflect the suggested flow.

---

> > ### Comment · Reviewer_bCcz · 2025-08-04
> >
> > Thank you for your rebuttal. The authors address my concern. I prefer to keep the score.

---

### Official Review · Reviewer_sZaS · 2025-07-03

**Clarity:** 3
**Significance:** 4
**Originality:** 2
**Rating:** 4
**Confidence:** 2

**Summary:**

This paper introduces MAESTRO, a new method for learning from time series data collected by multiple sensors (modalities), even when some sensors are missing. It uses symbolic tokens to represent data, gives different attention to each sensor based on its importance, and connects different sensors using efficient sparse attention. It also uses a Mixture-of-Experts system to adapt to different combinations of available sensors. MAESTRO is tested on four real-world datasets and outperforms 10 existing methods, showing strong results even when 40% of the sensor data is missing.

**Questions:**

1. The paper introduces a modality-aware attention budget mechanism to adaptively control sparsity per modality. Could the authors clarify how the gating network G is trained? Is there any regularization or inductive bias to prevent overfitting, especially when modality availability varies widely across samples?

2. The symbolic SAX tokenization is presented as a key component to reduce sequence length and encode missingness. Could the authors provide more quantitative or qualitative insights into why this representation improves performance (e.g., beyond Table 2 and Figure 3)? Would the model still benefit from SAX if trained on high-resolution sequences with sufficient compute?

3. The paper demonstrates robustness under missing modalities, but it is unclear whether MAESTRO can handle asynchronous modalities (e.g., with misaligned sampling rates and no temporal sync). Could the authors discuss how the current design generalizes to this setting, and whether additional temporal alignment strategies would be needed?

4. While MAESTRO uses sparse attention and MoE to improve efficiency, could the authors discuss performance under strict compute budgets (e.g., edge devices)? How does inference latency scale with modality count and sequence length, especially in high-frequency sensor settings?

**Ethical Concerns:**

["NO or VERY MINOR ethics concerns only"]

**Final Justification:**

Thank authors for the additional explanations and experiments, including those on edge devices. I want to maintain my score.

**Limitations:**

Yes. The authors have done a commendable job addressing the limitations of their work.

**Quality:**

3

**Strengths And Weaknesses:**

Strengths: (1) The paper presents a high-quality and well-designed framework. MAESTRO integrates multiple components—symbolic tokenization, adaptive attention budgeting, sparse attention, and MoE routing—into a cohesive pipeline that is robust and efficient. (2) The paper is clearly written, with intuitive explanations, detailed methodology, and helpful visualizations (e.g., Figures 2, 4, 5). The ablation studies and complexity comparisons further support the model design. (3) MAESTRO addresses a highly practical and underexplored challenge in multimodal time series learning: handling dynamic and incomplete sensor data. The ability to work under arbitrary modality combinations is highly relevant for real-world sensing systems. (4) The combination of symbolic time-series representation with adaptive attention and sparse MoE for multimodal fusion is novel.

Weaknesses: (1) While the method is well explained overall, the symbolic tokenization step and adaptive attention budgeting could benefit from more intuitive, real-world analogies or examples for broader accessibility. (2) While the combination is novel, many individual components (e.g., SAX, MoE, sparse attention) are adaptations of known ideas; the contribution lies more in system design than theoretical innovation.

---

> ### Author Rebuttal · Authors · 2025-07-31
>
> Thank you for your encouraging review and constructive feedback. We appreciate your recognition of the practical significance of our work and the merit in MAESTRO's design for real-world sensing applications. Please find our response to your questions below:
>
> > Could the authors clarify how the gating network 𝐺 is trained?
>
> The gating network, $G$, is composed of two feed-forward layers, and the output is the sigmoid of the last layer which produces an $M$-dimensional logit vector (where $M$ is the number of modalities). We train this gating network using the cross-entropy loss (given in lines 280-281) with the end-to-end MAESTRO. We do not use any additional regularization loss. During training, we follow a curriculum learning approach (as mentioned in lines 253–257), which increases the modality dropout monotonically over the course of training epochs, as shown in Figure 9 of the Appendix. This training strategy helps prevent overfitting to any specific modality combinations.
>
> Additionally, in our initial design iterations, we considered a loss term—sparsity loss—where we aimed to minimize the overall attention budget. The sparsity loss was constructed by taking the sum of the $M$-dimensional logit vector. The final loss considered was the cross-entropy loss and the sparsity loss. However, this did not result in stable training without constraining the sparsity loss. In future work, we will consider proactively sparsifying the attention budget based on the available computational resources.
>
>
> > Could the authors provide more quantitative or qualitative insights into why SAX representation improves performance (e.g., beyond Table 2 and Figure 3)? Would the model still benefit from SAX if trained on high-resolution sequences with sufficient compute?
>
>
> The key advantages of discrete binning achieved using SAX specifically for MAESTRO are: 1) the ability to reserve a symbol to naturally encode missingness, and 2) to reduce sequence length.
>
> An additional benefit of SAX—particularly for predictive tasks where discriminative patterns across the complete sequence inform task-specific predictions—is that mapping to a fixed vocabulary helps suppress local noise or spurious perturbations. This allows the model to focus on high-level trends that are more relevant to the task. Many previous works [1, 2, 3] have reaped this advantage and reported improved classification performance.
>
> In our case, to demonstrate the benefit of SAX tokenization, we conduct following two experiments.
>
> **Experiment I**: Using the WESAD dataset (stress detection with 5 modalities and 3 classes), we trained on high-resolution data without reducing sequence length by setting the word length (w) to 1. The results presented below confirm the benefit of SAX representation.
>
>
> | Tokenization Method       | Sequence Length <br>(after tokenization) | Accuracy | Std. Dev |
> |---------------------------|------------------------------------------|----------|----------|
> | SAX, w=2        | 128                                      | 0.77     | 0.02     |
> | SAX, w=1        | 256                                      | 0.75     | 0.04     |
> |Raw input (No SAX)                    | 256                                      | 0.69     | 0.02     |
>
>
> **Experiment II**: We add small Gaussian noise to 20% of the modalities (in this case, the wrist BVP, TEMP, and EDA) and conduct zero-shot inference. As expected, when the word length is set to 2, the effect of this small perturbation is neglible. Even with word length 1, the performance is much more robust compared to raw input
>
> | Tokenization Method | Clean Data | 20% Noisy Channels |
> |---------------------|------------|--------------------|
> | SAX, w = 2          | 0.77       | 0.75               |
> | SAX, w = 1          | 0.75       | 0.68               |
> | Raw input (No SAX)  | 0.70       | 0.51               |
>
> We will add these experiments to our Appendix section and refer to them in Section 3.2.1 to give our readers more intuition about SAX-based tokenization. Thank you for asking about this aspect.
>
> We would also like to draw your attention to Figure 5 and Section E.1 in the Appendix, which demonstrate that under arbitrary missingness, the proposed MAESTRO with SAX encoding outperforms its counterpart without SAX encoding.
>
> [1] Symbolic representation for time series, 2024
>
> [2] Bake off redux: a review and experimental evaluation of recent time series classification algorithms, 2024
>
> [3] SAX-VSM: Interpretable Time Series Classification Using SAX and Vector Space Model, 2013
>
> > Could the authors discuss how the current design generalizes to the settings of asynchronous modalities (e.g., with misaligned sampling rates and no temporal sync), and whether additional temporal alignment strategies would be needed?
>
> In the current work, we focus on handling complete modality missingness, which is a common issue in real-world sensing applications with a large number of modalities (N > 4). This typically occurs when a sensor fails and we are left with an arbitrary subset of multimodal inputs for prediction, as described in our paper's scope (lines 10, 62, 74).
>
> While a comprehensive study evaluating MAESTRO on irregularly sampled, asynchronous time-series is out of the scope of the current paper, we present a brief pilot study in a simplified setting. We mask approximately 25% of the samples from the beginning and end of the sequence in 20% of the modalities(in this case, the wrist BVP, TEMP, and EDA) during inference to demonstrate zero-shot transfer capability. Currently, the model treats this partial missingness similarly or slightly better than complete missingness.
>
>
> | Modalities                                      | Accuracy |
> |------------------------------------------------|----------|
> | All available                                  | 0.77     |
> | Completely missing 20% modalities              | 0.61     |
> | Mask first 25% time steps in 20% modalities    | 0.60     |
> | Mask last 25% time steps in 20% modalities     | 0.65     |
>
>
> This performance can potentially be improved by incorporating standard techniques such as absolute positional encoding, or employing prior models like mTAN [4] (multi-time attention network), which can better encode irregular time-series with semantic meaning to support downstream cross-modal learning. We will include this discussion and pilot experiment in our Appendix, and plan to explore these directions in the future. Thank you for commenting on this.
>
> [4] Multi-Time Attention Networks for Irregularly Sampled Time Series, 2021
>
>
> > Could the authors discuss performance under strict compute budgets (e.g., edge devices)? How does inference latency scale with modality count and sequence length, especially in high-frequency sensor settings?
>
> We conduct two experiments to demonstrate 1) MAESTRO's performance across two computation platforms and 2) benefit of sparse attention with simulated high sampling rate data stream.
>
> **Experiment I**: We implement MAESTRO with different number of experts and varying word lengths (which in turn reduce the sequence length), and report the accuracy, GFLOPs, and latency on two compute platforms—RTX A6000 and Jetson TX2.
>
> ##### Varying Experts for MoE (default w = 2)
>
>
> | # Experts | ACC_eval | RTX A6000 - Latency (ms) | RTX A6000 - GFLOPs  | Jetson TX2 - Latency (ms)   | Jetson TX2 - GFLOPs |
> |-----------|----------|--------------------------|---------------------|-----------------------------|---------------------|
> | 8		    |   0.80   |     352.09	              |       7.15          |	 1298.52	              |      6.16           |
> | 4		    |   0.79   |     279.41	              |       7.08          |	 1315.08	              |      6.12           |
> | 2		    |   0.75   |     245.03	              |       7.07          |	 1291.25	              |      6              |
>
>
> ##### Varying Word Length for SAX (default w = 2)
>
> | Word Length (w) | ACC_eval | RTX A6000 - Latency (ms) | RTX A6000 - GFLOPs | Jetson TX2 - Latency (ms) | Jetson TX2 - GFLOPs |
> |------------------|----------|---------------------------|----------------------|-----------------------------|-----------------------|
> |2		           |  0.80    |  279.41                   |	  7.08	             |        1315.08	            |       6.12           |
> |4		           |  0.77    |  270.00                   |	  3.56	             |        677.81	             |      3.06           |
> |8		           |  0.80    |  275.69                   |	  1.74	             |        349.33	             |      1.5           |
>
>
> **Experiment II**: We resample the modalities in WESAD to increase the sequence length in order to emulate high-frequency sensor readings and report the performance below. As expected, our proposed MAESTRO performs consistently even with longer sequence lengths. However, upon replacing the sparse attention with canonical dense self-attention layers in both the intra-modal and cross-modal stages, we encounter out-of-memory (OOM) issues at higher sampling rates (in this case, 128 Hz), highlighting MAESTRO’s advantage in resource efficiency. All experiments were conducted on the RTX A6000.
>
>
> | Sampling Rate | MAESTRO - Acc | MAESTRO - GFLOPs | MAESTRO - MMACs | Dense Attention - Acc | Dense Attention - GFLOPs | Dense Attention - MMACs |
> |---------------|----------------|-------------------|------------------|-------------------------|---------------------------|--------------------------|
> | 32            | 0.77           | 6.13              | 3066             | 0.75                    | 8.78                      | 4205                     |
> | 64            | 0.77           | 14.41             | 7205             | 0.71                    | 23.02                     | 11510                    |
> | 128           | 0.73           | 29.01             | 14502            | OOM                     | OOM                       | OOM                      |

---

> > ### Comment · Area_Chair_dYNK · 2025-08-07
> > **Follow-up discussion**
> >
> > Dear Reviewer sZaS,
> >
> > Thanks for reviewing the paper. Even though your initial rating was positive, we still want to hear your thoughts after rebuttal and if your opinion about the paper changes or remains the same after reading the rebuttal?
> >
> > Thanks, AC

---

> > ### Comment · Reviewer_sZaS · 2025-08-08
> >
> > I thank the authors for providing additional results and further clarifications. I will keep my score unchanged.

---

### Official Review · Reviewer_hWHy · 2025-07-04

**Clarity:** 3
**Significance:** 3
**Originality:** 4
**Rating:** 5
**Confidence:** 4

**Summary:**

The authors propose a new model (tokenization + encoding + prediction) scheme for multi-variate multi-modal time series that handles missingness and variable length sequences. The method tokenizes everything into discrete token sequences and then uses sparse attention mechanisms to deal with the long sequence length that results from concatenating the representations of multiple modalities. Experimentally, the authors compare 11 methods on 4 public datasets that physiological/health measurements.

**Questions:**

section 3, maestro.

- In the signal tokenization:

  - first step is clear: average each signal in windows

  - next step is unclear. "Similar to Lin et al. [28], we use equidistant Gaussian breakpoints to partition the range into α equiprobable regions, mapping each xˆ[w] to a symbolic token s[w] ∈ {s1, . . . , sα}.". What does equiprobable mean here? What's a region? How does one map x[w] to a choice of s[w]? i.e. what's the function phi? It feels not informative enough to cite another work and not mention what's actually done inline in thetext since it causes the reader to open another paper to understand a key detail. Skipping past it as a reader and just noting to self that there is somehow a tokenization that happens. It's referred to as a "symbolic regression" so I assume there is some function phi trained *somehow*.

- sparse self attention

  - Try to avoid introducing a mathematical symbol without defining it. For example, a(m_i, theta_a) is introducdd in line 195 in 3.2.2. but m isnt defined until the next section, which causes the reader extra mental burden in the earlier section.
  - Again, the "max-mean sparisty metric" from Zhou et al [63] used to compute the sparse self attention seems like a key detail but there's not even a basic math expression for it, instead it's delegated to appendix + citation. Like the earlier example with tokenization, this makes the details of the paper hard to follow as a reader must scroll up and down and open other papers to know what's going on.


Results

  - the main table 2 includes with and without SAX, and the ablation includes full vs sparse attention (two kinds). Was the ablation done with or without SAX? Also, what does full vs all-full attention mean?

**Ethical Concerns:**

["NO or VERY MINOR ethics concerns only"]

**Final Justification:**

My positive opinion of the work remains unchanged.

**Limitations:**

Yes.

**Paper Formatting Concerns:**

None.

**Quality:**

3

**Strengths And Weaknesses:**

Strengths:
- good connections with previous work
- consideration for both tokenization and prediction challenges in multivariate multi-modal time series
- attention to practical details like missingness
- attention to computational details like flops imposed by the architectural changes
- good benchmarking and many baselines run as far as dataset/model combos

Weaknesses:
- mostly writing clarity. A lot of complicated moving pieces and some key ones are deferred to references and appendices. See "Questions" below.

---

> ### Author Rebuttal · Authors · 2025-07-31
>
> Thank you for the encouraging review and the constructive feedback on improving our manuscript. We provide our response below :
>
>
> > Provide inline explanation of the symbolic tokenization instead of citing original reference for better readability.
>
> Thank you for the suggestion. We will add in the revision the following description of the breakpoint generation and mapping in the main text for easy referencing of the reader :
>
> "*To ensure an equiprobable distribution of symbols, we follow the empirical observation by Lin et. al [28] that normalized time series sequences follow a Gaussian distribution. We partition this distribution into $\alpha$ equal-sized areas under the Gaussian curve, referred to as regions, using breakpoints $\{\beta_0, \beta_1, \ldots, \beta_\alpha\}$, where each region corresponds to a unique symbol. The normalized piecewise aggregated value $\hat{x}[w]$ is then mapped to a symbolic token $s[w] \in \{s_1, \ldots, s_\alpha\}$ based on the breakpoint interval in which it falls, denoted using the mapping function $\phi(\hat{x}[w])$.*"
>
> ---
>
> >  "max-mean sparsity metric" from Zhou et al [63] used to compute the sparse self attention seems like a key detail but it's delegated to appendix + citation.
>
> Thank you. We will move the max-mean sparsity metric details from Appendix B to the Section 3.2.2 to improve readability.
>
> ---
>
> > a(m_i, theta_a) is introduced in line 195 in 3.2.2. but m isnt defined until the next section, which causes the reader extra mental burden in the earlier section.
>
>
> We will reorganize our writing to define $\mathbf{m}_i$ in Section 3.2.2 and explicitly mention that the design of $a(\mathbf{m}_i; \theta_a)$ is detailed in the following Section 3.2.3. Here is the planned updated text from line 195 :
>
> "*.... with a key modification: the sparsity budget is adaptively controlled by the learned function $a(\mathbf{m}_i; \theta_a)$, where $\mathbf{m}_i \in \mathbb{R}^{1 \times M}$ is a logit vector indicating the presence of each modality for an input sample $s^j_i$. Details on training $a(\mathbf{m}_i; \theta_a)$ are provided in Section 3.2.3.*"
>
>
> ---
> > Clarification of the experimental setup in the ablation studies -- Were they done with or without SAX? Difference between full vs. all-full attention.
>
> We conduct two kinds of design deep-dives in Section 4.2 –
>
> 1) Ablation of the design components in Figure 6: The first group of bars shows the ablation without SAX and demonstrates that this tokenization leads to an 8.5% and 5% improvement under full and 40% missingness in modalities during inference, respectively. For the remaining components, we conduct experiments with SAX (as it is a part of MAESTRO unless explicitly mentioned otherwise).
>
> 2) Sparse attention-derived Computational efficiency of MAESTRO: All experiments in this section are conducted with SAX. In the first two rows after MAESTRO, ``Full-Attn (Per-Modal)`` and ``Full-Attn (Cross-Modal)`` refer to variants where only the per-modal or cross-modal components, respectively, are replaced with dense attention. ``All Full-Attention`` indicates that both the per-modal and cross-modal attention modules use dense attention mechanisms instead of the proposed sparse design. We will update in the revision the caption of Table 3 as follows :
>
>     "*Computational Complexity. In MAESTRO, sparse attention in the Per-Modal and Cross-Modal components is replaced by dense attention, referred to as Full-Attn (Per-Modal) and Full-Attn (Cross-Modal), respectively. Replacing all sparse attention components with dense attention is denoted as All Full-Attn*."

---

> > ### Comment · Area_Chair_dYNK · 2025-08-07
> > **Follow-up**
> >
> > Dear Reviewer hWHy,
> >
> > Thanks for reviewing the paper. Even though your initial rating was positive, we still want to hear your thoughts after rebuttal and if your opinion about the paper changes or remains the same after reading the rebuttal?
> >
> > Thanks, AC

---

### Official Review · Reviewer_ALza · 2025-07-13

**Clarity:** 3
**Significance:** 2
**Originality:** 2
**Rating:** 4
**Confidence:** 2

**Summary:**

The paper considers the following problem: given M modalities where each modality is a multi-dimensional time series, the objective is to learn a label function that maps from the M modality time-series space to the label space of size C. The main challenge considered is that some of the modalities may be missing, and the labeling function is expected to be robust over arbitrary missing modalities. To this end the paper proposes the following approach:
1. Aggregation: applying piecewise aggregation with a window size T/W compressing the sequence length from size T to W.
2. Symbolic tokenization: mapping the compressed representations  into a symbol from a range of alpha values. One symbol is reserved to indicate missingness. The mapping is performed in a way that preserves the approximate distances before and after tokenization (Cor 3.2).
3. Feature extraction: using an encoder for each modality, based on a sparse self attention mechanism.
4. Cross modal learning: achieved by concatenating features along the temporal dimension to form a unified sequence. To overcome the L^2 attention complexity (especially after concatenation with large sequence length), the paper uses sparse cross attention achieving a complexity of L log L.

**Questions:**

Could the authors please provide intuition on how the proposed algorithm overcomes the exponential growth of missingness patterns?

**Ethical Concerns:**

["NO or VERY MINOR ethics concerns only"]

**Final Justification:**

My impression remains positive about the paper, however, I keep my score at 4 since I recommend acceptance but would also be ok otherwise due to the lack of novelty as the algorithm mainly combines existing approaches.

**Limitations:**

yes

**Quality:**

3

**Strengths And Weaknesses:**

Strengths:
- The paper is well-written and easy to follow.
- The proposed method is robust to any missing modality pattern which is challenging as the number of missing modalities combinations grows exponentially with the number of modalities.
- The authors algorithm introduces multiple operations to reduce computational complexity including: aggregation to decrease length from T to W, sparse self attention, and sparse cross attention.
- The numerical results show big improvements over state of the art for multiple datasets.

Overall, my impression  on the paper is positive, however, due to the following concerns I recommend a borderline accept.
Weaknesses:
- My main concern is regarding the limited novelty of the paper. Even though the results are good, the algorithm and techniques only combine those of existing works.
- The algorithm seems adhoc without sufficient explanation on why such approach could overcome the exponentially many missingness patterns challenge.
- Minor: the algorithm contains several hyper-parameters (the size W, alpha, ...) without any theoretical guidance on how they can be tuned, which introduces a lot of overhead in training.

---

> ### Author Rebuttal · Authors · 2025-07-31
>
> Thank you for your encouraging feedback on our work's pragmatism, performance and robustness. We provide our response below :
>
> > Even though the results are good, the algorithm and techniques
> largely combine those of existing works.
>
> We acknowledge that prior advancements in the fields of data mining [1] and long-horizon time-series forecasting [2] have inspired us; but our contribution lies in adeptly leveraging these ideas to solve a different problem--learning from dynamic and heterogeneous multimodal time series. We propose adapting the sparse attention technique at inter- and intra-modal stages with a learnable attention budget (the gating function in Section 3.2.3); using the opportunity with discrete symbols in tokenization to reserve a symbol for missingness (Section 3.2.1); and incorporating Mixture-of-Experts (MoE) to allow for implicit expert specialization under arbitrary modality combinations (Section 3.2.5)—which significantly extend beyond direct combinations of existing techniques. Our ablation study in Figure 6 shows the effectiveness of these proposed techniques.
>
>
> Additionally, our work presents an alternate viewpoint of multimodal handling of multi-sensor data over the commonly adopted multivariate modeling, which is often chosen because the sensing modalities share the same (numerical) representation in the input space. Our design considers the heterogeneity in the sensors and proposes an effective multimodal learning framework, MAESTRO.
>
> ---
>
> > Justification of how the algorithm overcomes the many missingness patterns challenge.
>
> We conduct an ablation of all our proposed components under complete observation and under modality missingness to demonstrate their roles (in Figure 6 of Section 4.2). The key intuition is that cross-modal attention (which improves performance by 21%) allows each modality to adequately query the necessary modalities—including itself—based on their availability and relevance. The modality dropout augmentation (improves performance by 9%) and the modality position embedding (improves performance by 16%), which denote the ordering of the modalities, help facilitate this learning. Supporting cross-modal learning with modality-specific embedding and augmentation helps MAESTRO achieve consistently superior performance (an average improvement of 9%) over other baselines, even under 40% missingness.
>
> ---
>
> > Minor: The algorithm involves several hyperparameters (e.g., window size $W$, $\alpha$) without theoretical guidance for tuning, adding training overhead.
>
> Thank you for commenting on this aspect. In our current work we have not extensively fine-tuned our hyper-parameters (as shown in Table 5 of the Appendix), yet we are able to outperform the existing baselines under complete and arbitrary modality combinations. However, for application-specific scenarios, based on the available computational resources, these hyperparameters can be selected to optimize for performance and computational efficiency. We present a brief sensitivity analyses for number of experts and the word length for the WESAD dataset across two computational platforms below :
>
> ##### Varying Experts for MoE (default w = 2)
>
>
> | # Experts | ACC | RTX A6000 - Latency (ms) | RTX A6000 - GFLOPs  | Jetson TX2 - Latency (ms)   | Jetson TX2 - GFLOPs |
> |-----------|----------|--------------------------|---------------------|-----------------------------|---------------------|
> | 8		    |   0.80   |     352.09	              |       7.15          |	 1298.52	              |      6.16           |
> | 4		    |   0.79   |     279.41	              |       7.08          |	 1315.08	              |      6.12           |
> | 2		    |   0.75   |     245.03	              |       7.07          |	 1291.25	              |      6.00             |
>
>
> ##### Varying Word Length for SAX (default w = 2)
>
> | Word Length (w) | ACC | RTX A6000 - Latency (ms) | RTX A6000 - GFLOPs | Jetson TX2 - Latency (ms) | Jetson TX2 - GFLOPs |
> |------------------|----------|---------------------------|----------------------|-----------------------------|-----------------------|
> |2		           |  0.80    |  279.41                   |	  7.08	             |        1315.08	            |       6.12           |
> |4		           |  0.77    |  270.00                   |	  3.56	             |        677.81	             |      3.06           |
> |8		           |  0.80    |  275.69                   |	  1.74	             |        349.33	             |      1.50          |
>
>
> We will include this analyses in our Appendix.
>
> ---
>
> [1] Experiencing SAX: a Novel Symbolic Representation of Time Series, 2007
> [2] Informer: Beyond Efficient Transformer for Long Sequence Time-Series Forecasting, 2021

---

### Note · Authors · 2025-08-16

We are very grateful for the time and insightful feedback from the reviewers. In this final note, we summarize our contributions acknowledged by the reviewers and additional experiments (including some noted in the paper as future work, e.g., noisy/asynchronous modalities) that were carried out during the rebuttal below:

**Summary of Contributions:**
* All reviewers consistently and explicitly acknowledge that our proposed framework, MAESTRO, is practical due to (1) *robustness to missing modalities* and (2) design components like sparse and adaptive attention budget, Mixture-of-Experts (MoE), symbolic tokenization (SAX), etc., to improve *computational efficiency*, beyond performance improvements. Additionally, MAESTRO provides an alternative to conventional multivariate modeling by explicitly handling sensor heterogeneity as multimodal data.
* **Reviewer ALza** explicitly recognizes MAESTRO's substantial performance benefits over the state of the art for multiple datasets.
* **Reviewers hWHy and sZaS** acknowledge MAESTRO's novelty in overall system design and appreciate connections with previous works. **Reviewer sZaS** specifically mentions that "The combination of symbolic time-series representation with adaptive attention and sparse MoE for multimodal fusion is novel."

**Additional Experiments During Rebuttal to Address Concerns:**

1. Additional **hyperparameter sweeps** for MoE count and word length in SAX representations to address **Reviewers ALza and bCcz**'s comments.
2. **Experiments on Jetson TX2** to further expand our method's evaluation and address **Reviewer sZaS**’s question about exploring other compute platforms.
3. Experiment to demonstrate potential robustness to **noisy modalities** as requested by **Reviewer sZaS**. We explored additive Gaussian noise, random noise, and Gaussian noise with spikes to simulate powerline noise.
4. Experiment to demonstrate potential handling of **asynchronous modalities** to address the concern of **Reviewer bCcz** by injecting time-dependent noise, where MAESTRO shows better robustness than the vanilla transformer.
5. Further **intuition of SAX** as requested by **Reviewer sZaS**, showing the benefit of SAX with different word lengths using a simulated high-frequency and noisy sensor input.

Our reviewers are overall positive about our work’s significance, quality, and correctness, and have acknowledged that our response has successfully addressed their concerns.

---

### Decision · Program_Chairs · 2025-09-17

**Decision:**

Accept (spotlight)

**Comment:**

# Summary
This paper proposes MAESTRO, a novel approach for learning from multimodal time series with missing modalities.

# Strength
* The proposed method is robust to missing modalities.
* Strong improvements over state of the art for multiple datasets.
* A well-designed framework, which integrates multiple components into a cohesive pipeline.

# Weakness
* Limited novelty with ad-hoc algorithms

# Recommendation
AC reads all reviews and agrees that (i) the technical contributions of proposed method are great; and (ii) the solving problem is a highly practical and under-explored challenge in multimodal time series modeling and it will be useful in various applications. Therefore, AC recommends an acceptance.